



**Spring tropical cyclones modulate near-surface isotopic compositions of**
**atmospheric water vapour at Kathmandu, Nepal**
Niranjan Adhikari[1, 2], Jing Gao[1, 3,*], Aibin Zhao[1], Tianli Xu[1, 4] Manli Chen[1, 3],
Xiaowei Niu[1], Tandong Yao[1, 3]
*[1] State Key Laboratory of Tibetan Plateau Earth System, Resources and Environment, Institute*
*of Tibetan Plateau Research, Chinese Academy of Sciences, Beijing 100101, China*
*[2] University of Chinese Academy of Sciences, Beijing 100049, China*
*[3] Lanzhou University, Lanzhou 733000, China*
*[4]Kathmandu Centre for Research and Education,* Chinese Academy of Sciences *–Tribhuvan*
*University, Kirtipur 44613, Kathmandu, Nepal*
* Correspondence to: Jing Gao, E-mail: gaojing@itpcas.ac.cn
**Abstract**
The Arabian Sea (AS) and the Bay of Bengal (BoB) are the major part of the Indian
Ocean where cyclonic activities prevail each year, resulting in extreme precipitation events,
particularly during the pre-monsoon season. Despite the significance of cyclones in Nepal, no
studies have investigated their impact on the isotopic composition of atmospheric water vapour
($\delta^{18}O_v$, $\delta D_v$, and d-excess$_v$). Here, we present the results of continuous measurements of the
isotopic composition of atmospheric water vapour at Kathmandu from 7 May to 7 June 2021
during two pre-monsoon cyclone events, namely cyclone Tauktae formed over the Arabian Sea,
and cyclone Yaas formed over the Bay of Bengal. We observed a significant depletion of $\delta^{18}O_v$
and $\delta D_v$ during both cyclone events compared to before and after the cyclone events which was





attributed to changes in moisture sources (local vs. marine) as inferred from backward moisture
trajectories. The outgoing longwave radiation (OLR) and regional precipitation during cyclone
events together with the observed correlation between vertical velocity and $\delta^{18}O_v$ showed high
moisture convergence and heavy convection at and around the measurement site which caused
unusually depleted $\delta^{18}O_v$ during that period. Moisture convergence and convection were stronger
during cyclone Yaas which resulted in higher (lower) d-excess$_v$ ($\delta^{18}O_v$), compared to Tauktae,
possibly due to strong downdrafts during the cyclone-related rain events which can transport
vapour with higher (lower) d-excess$_v$ ($\delta^{18}O_v$) toward the surface. Our study reveals that tropical
cyclones that originated from the BoB and the AS modulate isotopic signals of near-surface
atmospheric water vapour considerably in Nepal. Hence caution should be made while
interpreting the isotopic variability during the non-monsoon season and the effect of cyclones on
the isotopic composition of precipitation and atmospheric water vapour. Our results shed light on
key processes governing the isotopic composition of atmospheric water vapour at Kathmandu
and may have implications for the paleoclimate reconstruction of tropical cyclone activity.
Keywords: Cyclones; Bay of Bengal; Arabian Sea; Isotopic composition of atmospheric water
vapour;  Convection; Moisture convergence; Kathmandu



## 1 Introduction

Although the Indian summer monsoon accounts for more than 80 % of annual rainfall in
Nepal, agricultural activities also crucially rely on precipitation in the spring season (also known
as the pre-monsoon season). Pre-monsoonal rainfall in Nepal is often associated with cyclonic
events that provide sufficient moisture for precipitation to support the timely planting of
monsoonal crops. Every year, cyclonic events over the North Indian Ocean result in extreme
precipitation events, particularly during the pre-monsoon season with less extreme events during
the post-monsoon season (Li et al., 2013). Previous studies have suggested that extreme
precipitation in Nepal is mostly fuelled by moisture from the Arabian Sea (AS) and the Bay of
Bengal (BoB) (Bohlinger et al., 2017; Boschi and Lucarini, 2019). High sea surface temperatures
and the westward movement of tropical cyclones formed over the Western Pacific result in
cyclones being formed over the BoB and AS (Mohapatra et al., 2016). The number of cyclones
in the AS has dramatically increased in recent years compared to the number of cyclones in the
BoB (Pandya et al., 2021). According to the International Best Track Archive for Climate
Stewardship (IBTrACS) project, in 2019 three cyclones originated in the BoB while five
cyclones originated in the AS. This increase in cyclone frequency in the AS may be due to a rise
in sea surface temperature which also lengthens the cyclone decay period (Li and Chakraborty,
2020). Usually, the impact of cyclones formed over the AS is restricted to the nearest coastal
regions. However, in recent years this appears to have changed as cyclones are forming back-to-
back over the AS and affecting the entire Indian subcontinent including surrounding regions,
likely due to AS warming leading to cyclone intensification (Li and Chakraborty, 2020). Cyclone
Tauktae has affected the livelihoods of people both near the coast and further inland during the
pre-monsoon season of 2021 (Pandya et al., 2021). The impacts of cyclone Yaas after cyclone





Tauktae were also felt in Nepal, where it triggered flooding and landslides in several parts of the
country (https://floodlist.com/asia/nepal-flood-landslide-may-2021/). As both cyclones hit in
short succession, this led to severe agricultural damage in several parts of India at a critical time
when farmers were preparing to sow their rice paddies ahead of the monsoon season
(https://reliefweb.int/organization/acaps). In Nepal, most of the damage due to Yaas was mostly
limited to the Terai regions which experienced intense and continuous rainfall
(https://kathmandupost.com/). At the same time, some hilly regions benefited from these
cyclone-induced rains, as they created favourable conditions for farmers preparing their
monsoonal crops. Moisture flux associated with cyclones generally extends over a large area and
causes moderate to heavy precipitation along the cyclone path and on the nearest land mass
(Chan et al., 2022; Rajeev and Mishra, 2022). Thus, it is essential to understand the influence of
these extreme rainfall events on atmospheric water vapour, which are in turn related to local
clouds and surface energy budgets, determining the amount of moisture available to plants.

Atmospheric water vapour is an important constituent of the hydrological cycle and

climate system (Saranya et al., 2017), mainly because of its impacts on solar radiation
absorption, cloud formation, and atmospheric heating (Noone, 2012). With global warming, the
amount of water vapour in the atmosphere is also expected to increase. This has created scientific
interest in a variety of fields to elucidate the impact of atmospheric water vapour on changing
moisture patterns (Hoffmann et al., 2005).

The isotopic composition of atmospheric water vapour ($\delta^{18}O_v$, $\delta D_v$, and d-excess$_v$)

contains comprehensive information about the hydrological cycle and the history of moisture
exchange (Noone, 2012; Payne et al., 2007; Risi et al., 2008; Worden et al., 2007). Several
studies have shown that the isotopic composition of atmospheric water vapour is an effective



indicator of cyclone activity (Munksgaard et al., 2015; Sun et al., 2022) including cyclone
evolution and structure (Lawrence et al., 2002). The atmospheric water vapour and precipitation
associated with tropical cyclones tend to have extremely depleted isotopic compositions
compared to monsoonal rain (Chen et al., 2021; Jackisch et al., 2022; Munksgaard et al., 2015;
Sánchez-Murillo et al., 2019), which may be due to the high condensation efficiency and
substantial fractionation associated with cyclones. A few studies found a systematic depletion of
heavy isotopes towards the cyclone eye (Lawrence et al., 2002, 1998; Lawrence and Gedzelman,
1996; Sun et al., 2022; Xu et al., 2019). For instance, studying the cyclone Shanshan on Ishigaki
Island, southwest of Japan, Fudeyasu (2008) observed that isotopic depletion in precipitation and
water vapour increased radially inward in the cyclone's outer region, likely due to a rainout
effect associated with condensation efficiency and the isotopic exchange between precipitation
and water vapour. A study conducted in north-eastern Australia during cyclone Ita in April 2014
highlighted the role of synoptic-scale meteorological settings in determining the isotopic
variability of atmospheric water vapour (Munksgaard et al., 2015). In Fuzhou, China, Xu et al.,
(2019) reported a significant depletion in typhoon rain $\delta^{18}O$ which was related to the combined
effect of large-scale convection, high condensation efficiency, and recycling of isotopically
depleted vapour in the rain shield area. Sánchez-Murillo et al., (2019) highlighted the role of
convective and stratiform activity as well as precipitation type and amount as the main
controlling factors of precipitation stable isotopes associated with tropical cyclones. The impact
of high stratiform fractions and deep convection on isotopic depletion in precipitation during
typhoon Lekima was confirmed by Han et al., (2021). These findings clearly demonstrate that
the processes that contribute to high-frequency shifts in the isotopic composition of precipitation
and atmospheric water vapour during tropical cyclones are still a matter of debate.



112 Although several studies have examined the isotopic variation of event-based

113 precipitation in Nepal (Acharya et al., 2020; Adhikari et al., 2020; Chhetri et al., 2014), there

114 remains a knowledge gap regarding the isotopic response of atmospheric water vapour during

115 cyclone events. Here, we present for the first time the evolution of the isotopic composition of

116 atmospheric water vapour ($\delta^{18}O_v$, $\delta D_v$, and d-excess) in Kathmandu during two pre-monsoon

117 cyclone events. Isotopic data were provided in 2021, stretching from one week before to one

118 week after the cyclone events. Although neither cyclone passed directly over Kathmandu, their

119 remnant vapour produced several days of rainfall over Kathmandu which enabled us to observe

120 changes in the isotopic composition of atmospheric water vapour at high temporal resolutions

121 and to evaluate the cause of such changes at daily and diurnal scales.

122 **2 Data and methods**

123 **2.1 Site description**

124 The Kathmandu station lies on the southern slope of the Himalayas ($27°42'$ N, $85°20'$ E)

125 at an average altitude of about 1400 m above sea level. Based on an 18-year-long record (from

126 2001 to 2018) (Figure 1), this region has an average annual temperature of about $19°$ C and

127 average annual precipitation of 1500 mm, with most of the rainfall occurring in the monsoon

128 season (June to September) (Adhikari et al., 2020). About 16 % of total annual rainfall in

129 Kathmandu occurs in the pre-monsoon season (March to May) with a corresponding mean

130 maximum (minimum) air temperature of $28°$ C ($13°$ C) and relative humidity (RH) of 67 %. The

131 total moisture flux (sum of zonal and meridional fluxes) during the pre-monsoon season is low

132 (<60 kg/m/s) as is specific humidity (~6 g/kg), which is associated with transport by westerlies.

133 The region receives about 78 % of its annual rainfall during the monsoon season with associated



mean maximum (minimum) air temperature of 29$^{\circ}$C (20$^{\circ}$C) and RH of about 82 %. Most of the
precipitation over Kathmandu during the monsoon period is due to the influx of humid air
masses from the BoB. Average post-monsoon (October and November) and winter (December to
February) RH is about 80 % and 78 %, respectively, with similar rainfall contributions (3 %)
during both seasons. The mean post-monsoon and winter mean maximum (minimum) air
temperature is about 25$^{\circ}$C (11$^{\circ}$C) and 21$^{\circ}$C (4$^{\circ}$C), respectively.

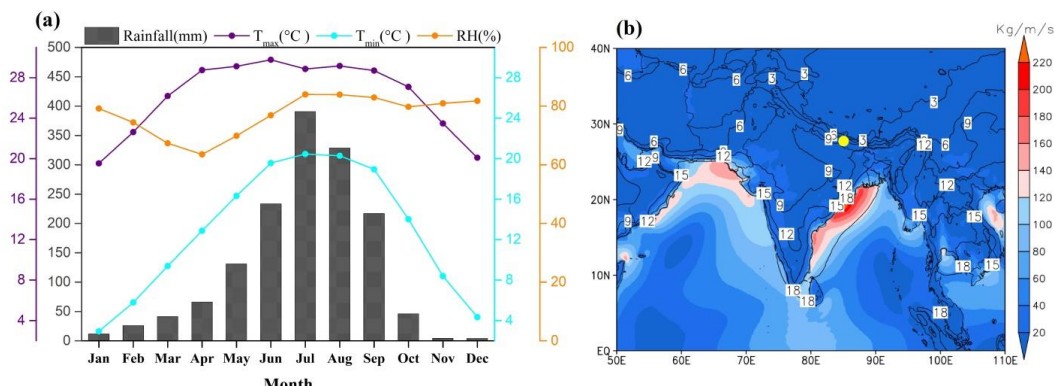


**Figure 1 (a) Long-term (2001-2018) average monthly maximum temperatures (T$_{max}$),**
**minimum temperature (T$_{min}$), relative humidity (RH), and precipitation amount (P) at**
**Kathmandu. (b) Spatial distribution of long-term (1990-2021) average specific humidity (in**
**g/Kg) (contour lines) and mean vertically integrated moisture flux (colour) during the pre-**
**monsoon season. The yellow dot shows the location of the Kathmandu site.**
**2.2   The evolution of cyclones Tauktae and Yaas**

Cyclone Tauktae developed as a tropical disturbance on 13 May 2021 over the AS and had

evolved into a deep depression by 14 May, moving northward and gradually intensifying over
the warm coastal water before turning into a cyclonic storm with wind speeds reaching 75 km/h
on that same day (Pandya et al., 2021).  Even after making landfall in the Gir-Somnath district of





Gujarat, Tauktae continued to strengthen and was classified as an extremely severe cyclonic
storm on 17 May reaching maximum wind speeds of 220 km/h as per the Indian Meteorological
Department's Tropical Cyclone Intensity Scale (Verma and Gupta, 2021; Pandya et al., 2021).
Cyclone weakened into a low depression on 18 May 2021 at 17:00 h Indian Local Time (ILT)
and finally dissipated one day later. Due to its large convective area, it brought heavy rainfall to
different regions of India and Nepal.
Cyclone Yaas started out as a depression over the BoB on 22 May 2021 at 08:30 h ILT
and gradually intensified into a deep depression before turning into a cyclonic storm on 24 May
at around 00:00 h ILT as it moved northeast (Paul and Chowdhury, 2021). The corresponding
wind speed and central pressure were recorded as about 65 km/h and 990 hPa, respectively. On
24 May at around 18:00 h ILT, it intensified into a severe cyclonic storm with wind speeds
ranging from 89 to 117 km/h before becoming a very severe cyclonic storm on 25 May at 12:00
h ILT with wind speeds from about 119 km/h to 165 km/h. It made landfall north of Odisha on
26 May with maximum sustained wind speeds of 130 km/h to 140 km/h and progressively
weakened into a depression on 27 May and dissipated over northern India on 28 May.
**2.3  Isotope measurements and meteorological data**
Near-surface $\delta^{18}O_v$ and $\delta D_v$ were measured continuously using a Picarro L2130-i
analyser based on wavelength-scanned cavity ring-down spectroscopy (WS-CRDS) (Brand et al.,
2009), located at the Kathmandu Centre for Research and Education (KCRE), established by the
Chinese Academy of Sciences with the collaboration of Tribhuvan University, Nepal. An inlet of
water vapour was placed 7 m above the grass-covered ground. The copper tube is heated using a
self-regulating heat trace isolated with armaflex. To prevent rain from being sucked into the tube,
the head of the inlet was covered with a plastic hood. A 10 L min$^{-1}$ pump quickly transported the



vapour from the inlet to the analyser. The automated standard delivery module (SDM) was used
for standard calibration. Each calibration was made with two reference standards that had been
calibrated against Vienna Standard Mean Ocean Water (VSMOW) covering the isotopic ranges
of ambient water vapour at Kathmandu. Each reference standard was measured continuously for
a total of 75 min each day at three different humidity levels (25 min per level). The evaporated
standard was then mixed with dry air obtained via Drierite™ desiccant (Merck, Germany) and
finally delivered to the Picarro analyser for isotopic measurements. The isotopic composition of
atmospheric water vapour is reported as parts per thousand (‰) relative to VSMOW using
$$\delta^* = (R_A / R_S - 1) \times 1000 \; [\text{‰}], \qquad\qquad (1)$$

where $\delta^*$ represents either $\delta D_v$ or $\delta^{18}O_v$, and $R_A$ and $R_S$ denote the ratios of heavy to light
isotopes ($^{18}O/^{16}O$ or D/H) in the sample and standard, respectively (Kendall & Caldwell, 1998;
Yoshimura, 2015). As suggested by Dansgaard, (1964), deuterium excess (d-excess$_v = \delta D_v - 8 \times \delta^{18-}$
$^{O}O_v$) is used as a tracer for moisture source conditions (Liu et al., 2008; Tian et al., 2001).  We
presented hourly isotopic composition of atmospheric water vapour between 7 May and 7 June
2021, covering the Tauktae and Yaas cyclone events (see previous section) including 1 week on
either side of the events. An automated weather station (AWS) continuously measured air
temperature, relative humidity, dew point temperature, wind speed and direction, rainfall
amount, surface pressure, etc. at a sampling rate of 1 min$^{-1}$.
**2.4  Cyclone track data**

The International Best Track Archive for Climate Stewardship (IBTrACS) project

containing best-track datasets of recent and historical tropical cyclones was used to obtain the
cyclone track data for this study (Knapp et al., 2010). We downloaded wind speeds, pressure,
and      cyclone      eye      location      information      (3-hour      resolution)      from



https://www.ncei.noaa.gov/products/. The latter was used to calculate the spatial distance
between the cyclone's eye and our measurement location.

## 2.5  Satellite precipitation and Outgoing Longwave Radiation data

We used Integrated Multi-satellite Retrievals for GPM (IMERG) from the Global
Precipitation Measurement (GPM) program with a spatial resolution of 0.1° latitude and
longitude to analyse the regional rainfall intensity before, during, and after the cyclone events,
following a previously reported method (Huffman et al., 2017). These high-resolution IMERG
data allow for the identification of convective rainfall areas and the passage of tropical cyclones
(Jackisch et al., 2022) and have been used previously to depict cyclone tracks and associated
rainfall intensities (Gaona et al., 2018; Jackisch et al., 2022; Villarini et al., 2011).
For outgoing longwave radiation (OLR), we used the National Centers for Environmental
Prediction (NCEP) daily reanalysis of datasets, with a spatial precision of 2.5° from longitude-
latitude grids (available at https://www.esrl.noaa.gov/psd/ (Kleist et al., 2009). OLR data has
already been used as an index of tropical convection (Liebmann and Smith, 1996). We further
obtained zonal and meridional wind, specific humidity, vertical velocity, vertical pressure, and
vertical distribution of relative humidity and temperature data from ERA5 datasets with a spatial
resolution of 0.25° from longitude-latitude grids (https://cds.climate.copernicus.eu/).

## 2.6  Moisture backward trajectory analysis

To assess the influence of moisture transport history on the isotopic composition of
atmospheric water vapour before, during, and after the cyclone events, we analysed 5-day
moisture backward trajectories that terminated at the sampling site using the Hybrid Single-
Particle Lagrangian Integrated Trajectory (HYSPLIT) model (Draxler and Hess, 1997). The



Global Data Assimilation System (GDAS) with a spatial resolution of 1° (Kleist et al., 2009) was
used to provide the meteorological forcing for the HYSPLIT model. Variations in specific
humidity along the moisture trajectories were also calculated. Since most of the atmospheric
vapour is contained within the bottom 2 km, we set the initial starting height for the moisture
backward trajectories to 500 m above ground. Additionally, using ERA5 datasets, we determined
the average boundary layer height at Kathmandu during the study period as about 620 m, which
confirms 500 m as an appropriate choice for the initial starting height to derive the moisture
trajectories.



## 3   Results and discussion

### 3.1   Water vapour isotope evolution before, during, and after cyclone events

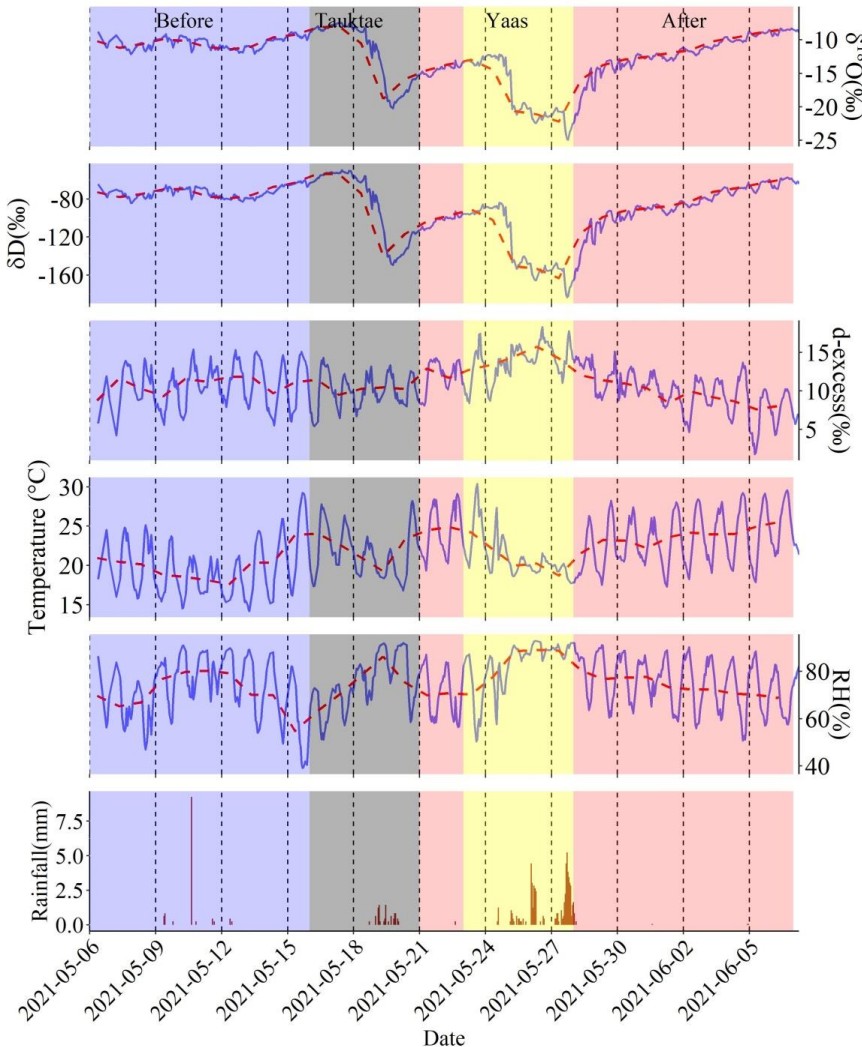

**Figure 2 Water vapour isotopic evolution (hourly averages) before, during, and after the Tauktae and Yaas cyclone events along with associated surface air temperature, relative humidity (RH), and rainfall amount. The red dashed line in the figure represents daily variations.**



The isotopic composition of atmospheric water vapour surrounding the two cyclone
events shows significant variability at Kathmandu station (Figure 2, Table 1). $\delta^{18}O_v$ and $\delta D_v$
showed a sudden depletion in the final stages of both cyclones, which coincides with RH
reaching its maximum values. The depletion was more pronounced during cyclone Yaas
compared to cyclone Tauktae. Before the cyclone Tauktae, $\delta^{18}O_v$ ($\delta D_v$) varied from -8.38 ‰ (-
60.10 ‰) to -12.10 ‰ (-84.15 ‰) with an average of -10.52 ‰ (-73.22 ‰) and d-excess$_v$ ranged
from 4.24 ‰ to 15.28 ‰ with an average of 10.94 ‰. The highest $\delta^{18}O_v$ value of -7.40 ‰ was
observed before the cyclone Tauktae, whereas the lowest $\delta^{18}O_v$ value of -24.92 ‰ was observed
during the final stages of cyclone Yaas. Clearly, the isotopic composition of atmospheric water
vapour shows a downward trend as the remnant of cyclones passed over Kathmandu. $\delta^{18}O_v$
decreased by over 12 ‰ from 14 May to 20 May (Tauktae) and again between 24 May and 29
May (Yaas), reaching minima for $\delta^{18}O_v$ ($\delta D_v$) of -20.21 ‰ (-149.49 ‰) and -24.92 ‰ (-183.34
‰), respectively. During Tauktae, d-excess$_v$ varied from 6.47 ‰ to 18.79 ‰ with an average of
10.87 ‰ while during Yaas it varied from 8.71 ‰ to 18.29 ‰ with an average of 13.77 ‰. After
both cyclones had dissipated, $\delta^{18}O_v$ (and $\delta D_v$) started to recover pre-cyclone values of -8.29 ‰ to
-14.94 ‰ (-57.40 ‰ to -109.31 ‰), with an average of -11.09 ‰ (-79.38 ‰). During that
period, d-excess ranged between 1.80 ‰ and 15.11 ‰ with an average of 9.37 ‰. Notably, the
isotopic composition of atmospheric water vapour before the commencement of rainfall by
Tauktae remained enriched, suggesting that the isotopic composition of atmospheric vapour
during that period was representative of surface layer inflow (Munksgaard et al., 2015).
However, $\delta^{18}O_v$ and $\delta D_v$ at the earlier stage of cyclone Yaas were significantly lower as
compared to the earlier stage of cyclone Tauktae. These discrepancies might be due to the timing
of their occurrence or the convective strength. The passage of cyclones that had formed over the



AS (Tauktae) and BoB (Yaas) caused significant depletion in the isotopic composition of
atmospheric water vapour and led to cumulative rainfall of 9.2 mm (Tauktae) between 14 May
and 20 May 2021 and 59.6 mm (Yaas) between 25 May and 28 May 2021 at our site. This is in
agreement with previous studies which documented similar depletion in isotope ratios due to
cyclone-associated intense rainfall (Krishnamurthy and Shukla, 2007; Rahul et al., 2016). It is
noteworthy that the above $\delta^{18}O_v$ minimum observed during cyclone Yaas is similar to the
minimum observed in Bangalore, India ($\delta^{18}O_v$ =-22.5 ‰) (Rahul et al., 2016) and Roorkee, India
($\delta^{18}O_v$ =-25.35 ‰) (Saranya et al., 2018) when cyclones evolved over the BoB, were closest to
their sampling sites. These results indicate the significant impact of oceanic moisture on the
isotopic composition of atmospheric water vapour over the continental during the time of
cyclones. We will discuss the influence of moisture sources in Sect. 3.3 in more detail.
The relation between $\delta^{18}O_v$ and $\delta D_v$ varies for the periods before, during, and after the
cyclones, showing different slopes and intercepts with the Local Meteoric Vapour Line (LMVL)
(Figure 3). Before the first cyclone event, both the slope and intercept are significantly lower
(slope=5.85 and intercept= -12.12), indicating the strong influence of non-equilibrium processes
such as evaporation. During both cyclone events, both the slopes and intercepts resemble the
slope and intercept of the global meteoric water line (GMWL: $\delta D=8\times\delta^{18}O+10$) (Figure 3). After
the cyclone events, the slope and intercept decreased to 7.37 and to 2.34, respectively, which
implies a change of moisture sources and evaporation becoming dominant once again.





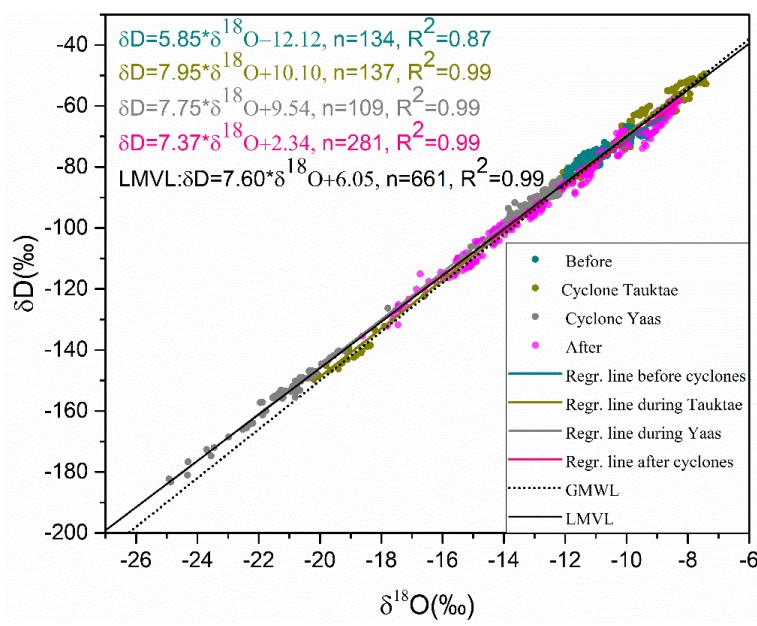

Figure 3 Relationships between $\delta^{18}O_v$ and $\delta D_v$ before, during, and after the cyclone events.
The regression lines for each period are presented along with GMWL for comparison.

Table 1 Descriptive statistics of $\delta^{18}O_v$, $\delta D_v$, and d-excess$_v$ measured before, during, and after the cyclone events.

| Period | $\delta^{18}O_v$ [‰] | | | $\delta D_v$ [‰] | | | d-excess$_v$ [‰] | | |
|---|---|---|---|---|---|---|---|---|---|
| | min | max | avg | min | max | avg | min | max | avg |
| Before | -12.10 | -8.38 | -10.52 | -84.15 | -60.10 | -73.22 | 4.24 | 15.38 | 10.94 |
| Cyclone Tauktae | -20.21 | -7.40 | -13.59 | -149.49 | -49.53 | -97.88 | 6.47 | 18.79 | 10.87 |
| Cyclone Yaas | -24.92 | -12.17 | -17.87 | -183.34 | -83.85 | -129.18 | 8.71 | 18.29 | 13.77 |
| After | -14.94 | -8.29 | -11.09 | -109.31 | -57.40 | -79.38 | 1.80 | 15.11 | 9.37 |





### 3.2 Day-to-day and diurnal variations during cyclones events


To understand the depletions in $\delta^{18}O_v$ and $\delta D_v$ during the Tauktae and Yaas cyclone
events, we analysed the regional wind fields and specific humidity over the Northern Indian
Ocean during the respective periods. Fig. S3 shows the genesis, development, movement, and
dissipation of cyclone Tauktae together with changes in specific humidity along the transport
path. The remnants of cyclone Tauktae caused light rain at Kathmandu, with a significant
depletion in $\delta^{18}O_v$ ($\delta D_v$) by ~8 ‰ (~66 ‰) on 20 May as compared to the previous day. From
the formation of a depression over the AS on 14 May 2021 until the commencing dissipation
inland on 19 May, no significant variation in the isotopic composition of atmospheric water
vapour was observed (Fig. 2). After the dissipation, when the residual Tauktae vapour passed the
Kathmandu producing light rains, $\delta^{18}O_v$ and $\delta D_v$ began to decrease independently of the rainfall
amount, starting on 19 May at around 11:00 h Local Time (LT) from -8.34 ‰ for $\delta^{18}O_v$ and -
56.06 ‰ for $\delta D_v$ and dropping in just one hour to -10.12 ‰ and -68.41 ‰ respectively. This
decrease continued for another day, reaching a minimum of -20.21 ‰ and -149.49 ‰ for $\delta^{18}O_v$
and $\delta D_v$ respectively on 20 May at around 12:00 h LT. However, d-excess$_v$ did not show notable
variations during the passage of cyclone Tauktae. $\delta^{18}O_v$ and $\delta D_v$ remained anomalously depleted
from 20 to 22 May due to the presence of a remnant of cyclone Tauktae.
On 24 May, cyclone Yaas formed over the BoB and started along a northward trajectory
through north-eastern India (Fig. S4). The high specific humidity over India and surrounding
regions during the days of cyclone formation indicates that Yaas had lifted a large amount of
water vapour from the BoB, which subsequently produced intense rainfall along its path. The
effect of cyclone Yaas on $\delta^{18}O_v$ and $\delta D_v$ at Kathmandu was first captured on 25 May with $\delta^{18}O_v$
($\delta D_v$) dropping rapidly from -12.62 ‰ (-88.71 ‰) on 25 May at 20:00 h LT to -15.07 ‰ (-



106.22 ‰) just one hour later. At the same time, d-excess$_v$ was increased from 12.30 ‰ to 14.34
‰. The depletion continued until 28 May with a minimum of $\delta^{18}O_v$ ($\delta D_v$) by -24.92 ‰ (-182.35
‰) at 16:00 h LT. At that time, Yaas had already weakened into a low-pressure area over Bihar
in south-eastern Uttar Pradesh, India. $\delta^{18}O_v$ and $\delta D_v$ started to increase after Yaas had dissipated,
reaching -14.64 ‰ for $\delta^{18}O_v$ and -103.97 ‰ for $\delta D_v$ on 29 May at 16:00 h LT. From 25 to 29
May, d-excess$_v$ gradually increased, resulting in a strong negative association with $\delta^{18}O_v$ and
$\delta D_v$, with correlation coefficients of -0.60 and -0.55 respectively. Such strong isotopic depletion
during cyclone events might be associated with high condensation efficiencies within the
cyclones leading to extensive fractionation (Rahul et al., 2016).
To further elucidate the processes affecting the diurnal variability of the isotopic
composition of atmospheric water vapour, we investigated the mean diurnal cycles of $\delta^{18}O_v$, $\delta D_v$,
d-excess$_v$, surface temperature, and specific humidity during the cyclone events, focussing on the
last 4 days of each cyclone (19 May to 22 May for Tauktae and 25 May to 28 May for Yaas)
when the measurement site received the first precipitation caused by cyclones. Surprisingly, we
observed very weak diurnal signals in $\delta^{18}O_v$ and $\delta D_v$ during either cyclone event (Figure 4), with
amplitudes of diurnal variations in $\delta^{18}O_v$ ($\delta D_v$) of 1.10 ‰ (10.21 ‰) during cyclone Tauktae and
2.06 ‰ (16.07 ‰) during cyclone Yaas. The surface temperature and specific humidity showed
an average peak-to-peak variability of about 7 °C and 2 g/kg, respectively, during the cyclone
Tauktae. In contrast, these values were considerably lower during Yaas with respective peak-to-
peak variabilities of about 3 °C and 0.94 g/kg. Unlike $\delta^{18}O_v$ and $\delta D_v$, d-excess$_v$ showed a clear
diurnal pattern consisting of a gradual increase from early morning till about midday, followed
by about 4:00 h during which d-excess remained at a high level, before starting to gradually
decrease from about 16:00 h onward (Figure 4). This diurnal variation in d-excess$_v$ seems to have



been more prominent during cyclone Tauktae with a peak-to-peak variability of 3.87 ‰ (vs 1.90
‰ during cyclone Yaas). The d-excess$_v$ diurnal cycle during Tauktae was strongly synchronized
with surface temperature and specific humidity with respective correlation coefficients ($R^2$) of
0.96 and 0.81. During Yaas, the synchronicity was considerably weaker exhibiting correlation
coefficients ($R^2$) of 0.27 and 0.35 with temperature and specific humidity, respectively.
Considering that rather smaller precipitation amount during Tauktae compared to Yaas, neither
$\delta^{18}O_v$ nor $\delta D_v$ showed any notable diurnal signal during these events, indicating that any diurnal
variation in $\delta^{18}O_v$ or $\delta D_v$ during the cyclones events was independent of the day-night variation
in local weather parameters and the Rayleigh fractionation processes they underwent during their
northward movement (see Sect. 3.3 for a more detailed discussion); whereas local weather
parameters may play pronounced roles on d-excess$_v$ diurnal variations depending on rainfall
strength.




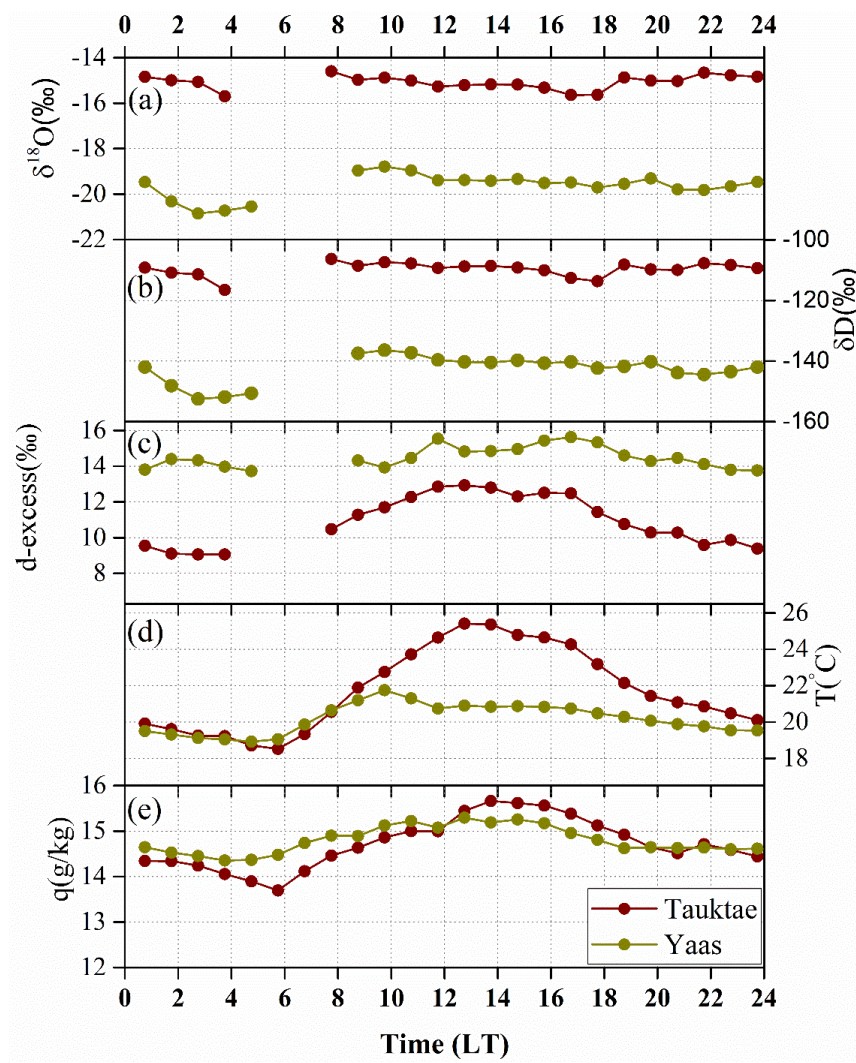


**Figure 4 Diurnal cycles of (a) $\delta^{18}O_v$, (b) $\delta D_v$, (c) d-excess$_v$, (d) temperature (T), and (e)**

**specific humidity (q) averaged from 19 to 22 May during Tauktae and from 25 to 28 May**

**during Yaas. The units of Time "LT" indicates Local Time.**


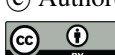



### 3.3 Isotopic response to regional climate

We now probe the underlying reasons for these isotopic variations in more detail. For this purpose, we analysed the influence of moisture sources on the isotopic composition of atmospheric water vapour by calculating 5-day backward trajectories for each day before, during, and after the cyclone events. We also calculated the associated specific humidity along the cyclone trajectories to estimate moisture uptake and identify possible rainfall regions (Figure 5).

Before the cyclone events, the majority of moisture trajectories associated with high $\delta^{18}O_v$ and $\delta D_v$ originated either locally or were brought in by westerlies with low specific humidity along their paths. During Tauktae, most trajectories originate in the AS. During Yaas, most trajectories point to the BoB as the sole vapour source contributing to the moisture at the sampling site (Figure 5).

Both cyclone events have in common that the specific humidity tends to be high while they are over oceans and the air becomes drier while crossing over land, as moisture is removed through precipitation. We found that the association between both $\delta^{18}O_v$ and $\delta D_v$ and Temperature/Relative humidity was much stronger during the cyclone events compared to before or after the events (Table 2). This might be linked to the cyclones transporting large amounts of moisture from remote oceans (Chen et al., 2021; Xu et al., 2019). After the cyclones had dissipated, the isotopic composition of atmospheric water vapour reverted to the original (enriched) levels ($\delta^{18}O_v$ = -14.64 ‰, $\delta D_v$= -103.97 ‰, and d-excess$_v$= 13.20 ‰).




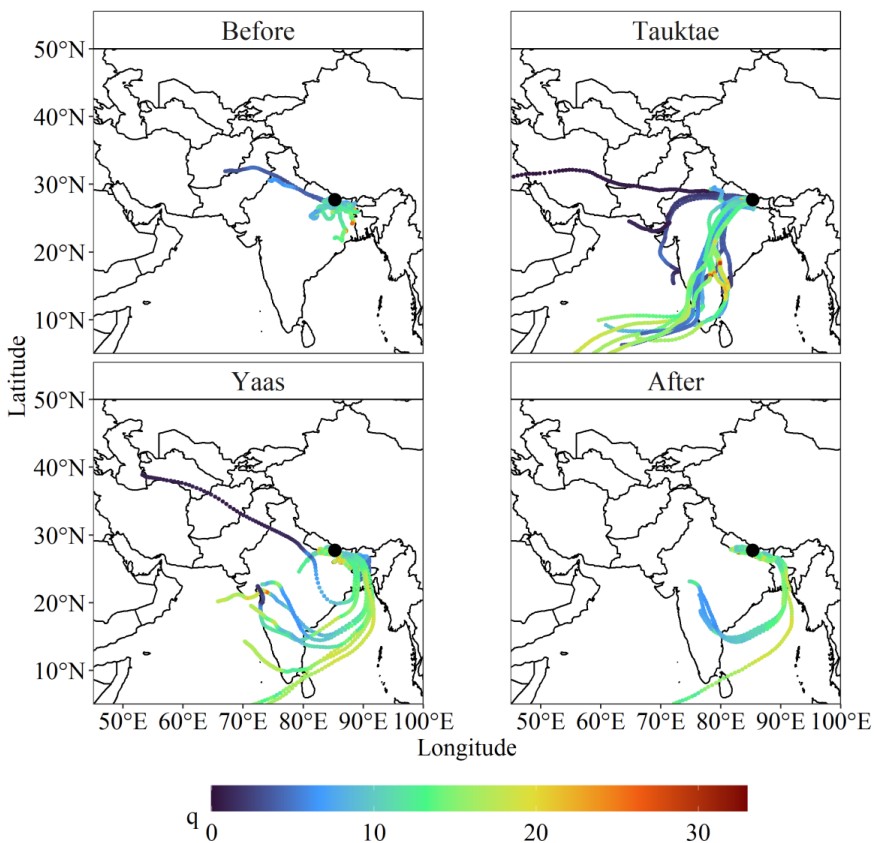

**Figure 5 Five-day backward moisture trajectories reaching the sampling site before, during, and after the cyclone events. Colours denote specific humidity (q in g/kg) along the trajectories.**

One of the most likely causes for large isotopic depletion during cyclone events might be the associated convection processes. Several studies have demonstrated that convective processes within tropical cyclones can cause the unusually depleted isotopic composition of precipitation and atmospheric water vapour (Fudeyasu et al., 2008; Jackisch et al., 2022; Munksgaard et al., 2015) due to a combination of strong cyclonic circulation, intense large-scale convection, heavy precipitation, and high wind speeds (Chen et al., 2021; Xu et al., 2019). Here



we analyse the relationship between the isotopic composition of atmospheric water vapour and
convective processes during two cyclone events, using outgoing longwave radiation (OLR) and
vertical velocity as a proxy for convection. Due to the frequent co-occurrence of intense
convection and significant mid-tropospheric convergence of moist air, the vertical velocities can
also serve as a proxy for convective activity (Lekshmy et al., 2014). Fig. S5 and Fig. S6 depict
the prevalence of strong convective processes associated with both cyclones throughout their
entire lifespans. During the initial days of cyclone formation, OLR exceeded 260 Wm$^{-2}$ in the
area of the sampling site (Figs. S5 and S6) and had rather decreased rapidly to below 200 Wm$^{-2}$
in the final stages of both cyclones when they were approaching the sampling site. Although the
amount of precipitation associated with Tauktae (9.2 mm) was much lower than with Yaas (59.6
mm), $\delta^{18}O_v$ depleted by up to 12 ‰ during both cyclone events. Importantly, during both
cyclones events, the progressive rainout was evident along the entire cyclone track, and the
spatial distribution of precipitation was highly correlated with the convective process (as
indicated by low OLR) (Figs. S7 and S8), suggesting that rainfall occurred from the deep
convective cloud rather than from local evaporation. This interpretation was confirmed by
comparing the regional precipitation to *in situ* measurements. According to Fig. S7 and Fig. S8,
the measurement site received its first rainfall on 19 May during cyclone Tauktae and on 25 May
during cyclone Yaas which we can confirm with our observation data. *In situ* observations show
that during the days leading up to cyclone Tauktae, the sampling site received a total of 12.2 mm
(from 7 May to 14 May) of precipitation with maximum rainfall of 9.2 mm/h recorded on 11
May at 13:00 h LT, which is equal to the total accumulated rainfall during the entire cyclone
Tauktae. Although the pre-Tauktae and during-Tauktae rainfall amounts are similar, pre-cyclone
$\delta^{18}O_v$ and $\delta D_v$ were significantly more enriched (averages:$\delta^{18}O_v$ = -11.83 ‰ and $\delta D_v$ = -80.30



‰) than during the cyclone event (averages: $\delta^{18}O_v$ = -13.59 ‰ and $\delta D_v$ = -149.49 ‰). This confirms our previously stated hypothesis that the rainfall associated with cyclones causes significantly lower isotope values in vapour due to intense convective system (Gedzelman et al., 2003; Kurita, 2013), which is absent in localized rain events and on days without precipitation (Lekshmy et al., 2022).

Our hypothesis that isotopic variations during cyclone events at Kathmandu are mainly driven by convective processes is further supported by the Hovmöller diagram of OLR averaged over 80-90° E (Figure 6), which clearly shows that $\delta^{18}O_v$ depletion coincides with the presence of clouds. In contrast, d-excess$_v$ showed rather dissimilar variations between both cyclone events. Before the arrival of cyclone Tauktae, the daily averaged d-excess was above the global average of 10 ‰ (Fig. 6, horizontal orange line). Once Tauktae was approaching the sampling site, d-excess$_v$ decreased from around 12 ‰ to 10 ‰ and continued to oscillate about 10 ‰ until Tauktae had dissipated. As cyclone Yaas approached the measurement site with intense rainfall (Fig. 2), d-excess$_v$ ($\delta^{18}O_v$) gradually increased (decreased) while RH increased and air temperature decreased (Fig. 2). More specifically, d-excess$_v$ on 24 May was recorded as 12.82 ‰ when surface air temperature and surface RH was about 24 $^{\circ}$C and 70 % respectively. On 27 May, we noticed about a 3 ‰ rise in d-excess$_v$ when the surface temperature was reduced by 4 $^{\circ}$C and the surface RH was increased by 19 %. The combination of increasing d-excess and decreasing $\delta^{18}O_v$ has also been observed during the active convective phase of Madden-Julian oscillations (MJO) in the tropical atmosphere which highlights the role of vapour recycling due to the subsidence of air masses from stratiform clouds (Kurita et al., 2011). In addition, a large increase in d-excess$_v$ was also recorded in atmospheric vapour during cyclone Ita in 2014 and was attributed to downward moisture transport above the boundary layer (Munksgaard et al.,



2015). In our case, we did not find any statistically significant correlation during cyclone Yaas
between d-excess$_v$ and RH/Temperature, although RH is generally considered an important
parameter for interpreting d-excess values in atmospheric vapour and precipitation (Pfahl and
Sodemann, 2014; Steen-Larsen et al., 2014). The observed co-occurrence of higher d-excess$_v$,
lower temperatures, and high relative humidity (Fig. 2) points to kinetic fractionation processes
either at a larger scale or in association with downdrafts (Conroy et al., 2016). This relationship
also highlights the role played by the convective process with regard to the isotopic composition
of atmospheric water vapour. Low $\delta^{18}O_v$ in combination with high d-excess$_v$ are known to be
associated with rain re-evaporation under conditions of high saturation deficit because the
addition of re-evaporated vapour to the atmosphere during precipitation events produces depleted
cloud vapour and high d-excess (Conroy et al., 2016; Lekshmy et al., 2014). On normal days
(without cyclones), high d-excess$_v$ values were generally accompanied by low RH (Figure 7) and
vice versa. However, high relative humidities of the surface air together with near saturation
conditions vertically (Figure 8, middle panel) during cyclone Yaas, rule out any effect of re-
evaporation on increased (decreased) d-excess$_v$ ($\delta^{18}O_v$ and $\delta D_v$) values. Hence, we surmise that
the higher d-excess$_v$ values during cyclone Yaas might be associated with downdrafts during
convective rain events, which can transport isotopically depleted vapour with higher d-excess$_v$
values from the boundary layer to the surface (Kurita, 2013; Midhun et al., 2013).





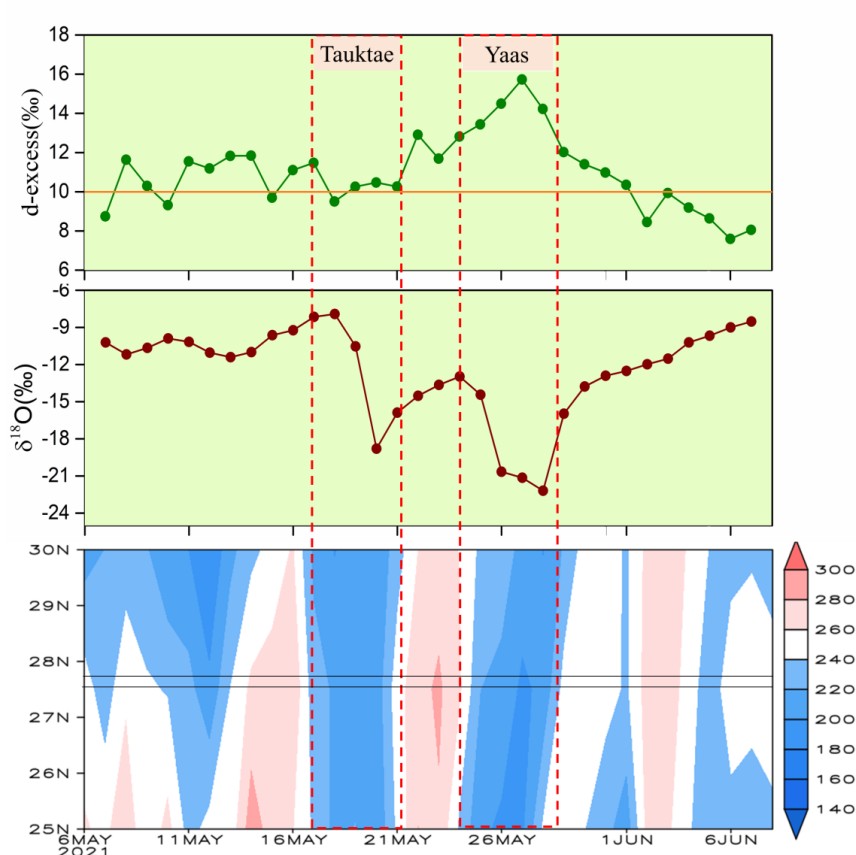


**Figure 6 Time series of daily averaged d-excess$_v$ (top panel), $\delta^{18}O_v$ (middle panel), and Hovmöller diagram of NOAA interpolated OLR (W/m$^2$) averaged over 80° E-90° E (bottom panel) The orange horizontal line in the top panel represents the global average d-excess value (i.e. 10 ‰) and solid parallel lines in bottom panel depict the latitude range of sampling site.**



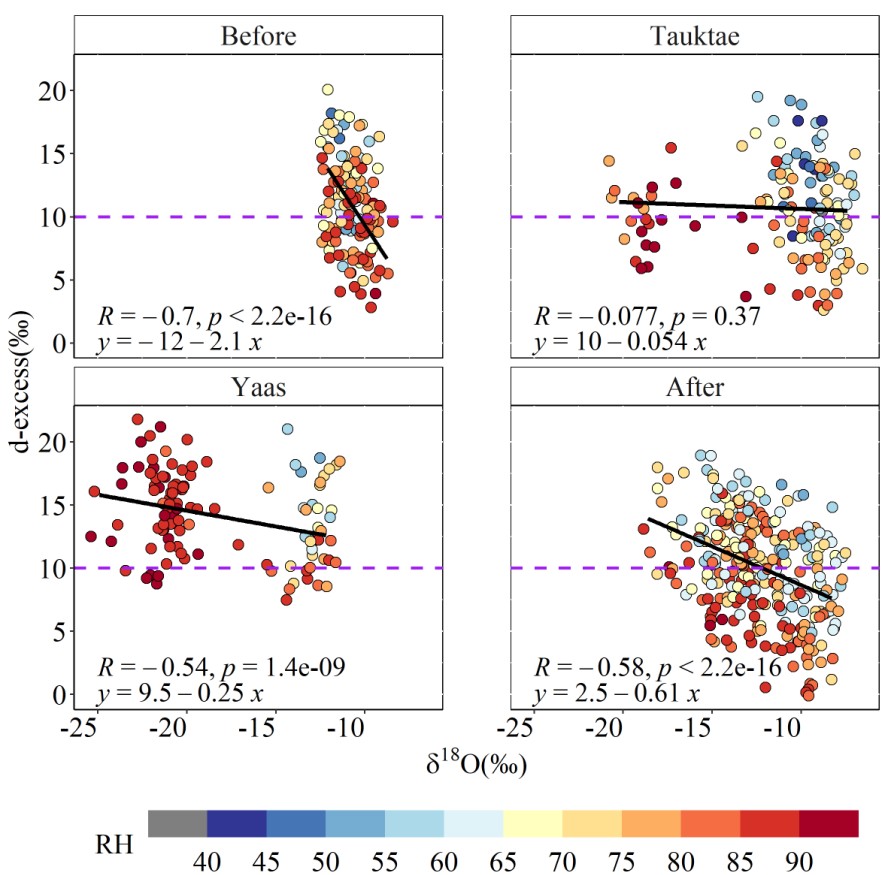


**Figure 7 Scatter plots of d-excess$_v$ vs. δ$^{18}$O$_v$ before, during, and after the cyclone events.**

**The colour represents RH (in %) and the horizontal dashed purple lines represent the**

**global average d-excess value (10 ‰).**

To further elucidate the impact of convection on the isotopic composition of atmospheric
water vapour, we analyzed the vertical distribution of vertical velocity, relative humidity, and air
temperature averaged over a box between 25° N-28° N and 83° E-87° E which has our
measurement site near its centre (Fig. 8). Our results show that strong shifts in δ$^{18}$O$_v$, δD$_v$, and d-
excess$_v$ during the cyclone events were strongly associated with vertical air motions (Figure 8).



We observed a general downward movement of air before the commencement of rainfall by
Tauktae (i.e., from 7 May to around 18 May). The high depletion of $\delta^{18}O_v$ and $\delta D_v$ during the
final stages of Tauktae (Figure 2) was accompanied by strong upward air movement extending
from 800 hPa to about 200 hPa (Figure 8). This upward motion was even stronger during
cyclone Yaas and already became evident near the measurement site once Yaas made landfall at
the BoB coast on 26 May. Interestingly, variations in RH at different pressure levels strongly
coincided with changes in vertical velocity while the lower troposphere remained near saturation
(RH= ~100 %) during the late stages of both cyclones. While the vertical air temperature showed
the expected progressive decline with altitude, there were no significant temporal variations in
temperature during the entire period, despite the high variation in RH. This implies that the high
RH in the lower troposphere during both cyclone events was independent of temperature and
hence the result of deep convection and the widespread development of clouds. The strong
convective updraft added additional moisture from the warm ocean below, before passing over
our measurement site (Lekshmy et al., 2014). Convective updrafts cause moisture to condense
quickly and this high-efficiency condensation of heavy rain can result in more depleted $\delta^{18}O_v$
and $\delta D_v$ (Lawrence and Gedzelman, 1996). In addition, we found a strong positive correlation
between $\delta^{18}O_v$ and average vertical velocity (r=0.57) during Yaas at pressure levels between 300
hPa and 600 hPa in the area surrounding our study site (cf., Lekhsmy et al., 2014); During
Tauktae, this correlation was weaker but still significant (r=0.30) (Fig. S9). This result suggests
that the higher depletion in $\delta^{18}O_v$ and $\delta D_v$ during cyclone Yaas relative to Tauktae may be due to
the stronger convection associated with the BoB vapour compared to the AS vapour. The BoB is
a convectively active region, and previous studies reported greater depletions in $\delta^{18}O_v$ and $\delta D_v$ in
precipitations with moisture from the BoB compared to the AS, irrespective of the season



(Breitenbach et al., 2010; Lekshmy et al., 2015; Midhun et al., 2018). Another reason why we
observed different levels of isotope depletion between both cyclones may be related to
differences in their closest proximity to the sampling site. While Yaas came as close as 400 km
to our study site, Tauktae was still 1100 km away when it dissipated (Fig. S10). The closer
proximity of Yaas may explain the stronger rainfall during that event which enhanced the
isotopic fractionation which in turn led to stronger isotopic depletion (Jackisch et al., 2022).
Similar results during the cyclone events have already been documented for precipitation stable
isotopes (e.g., Fudeyasu et al., 2008; Jackisch et al., 2022; Munksgaard et al., 2015; Xu et al.,
2019) and water vapour stable isotopes (e.g., Munksgaard et al., 2015; Rahul et al., 2016;
Saranya et al., 2018). Even after both cyclones had dissipated, progressive rainfall continued at
our sampling site due to the presence of residual moisture from the cyclones. Once these residual
effects had diminished and rainfall intensity weakened, did both $\delta^{18}O_v$ and $\delta D_v$ start to increase
again (Fig. 2), likely due to evaporative effects (Munksgaard et al., 2015; Xu et al., 2019;
Jackisch et al., 2022).

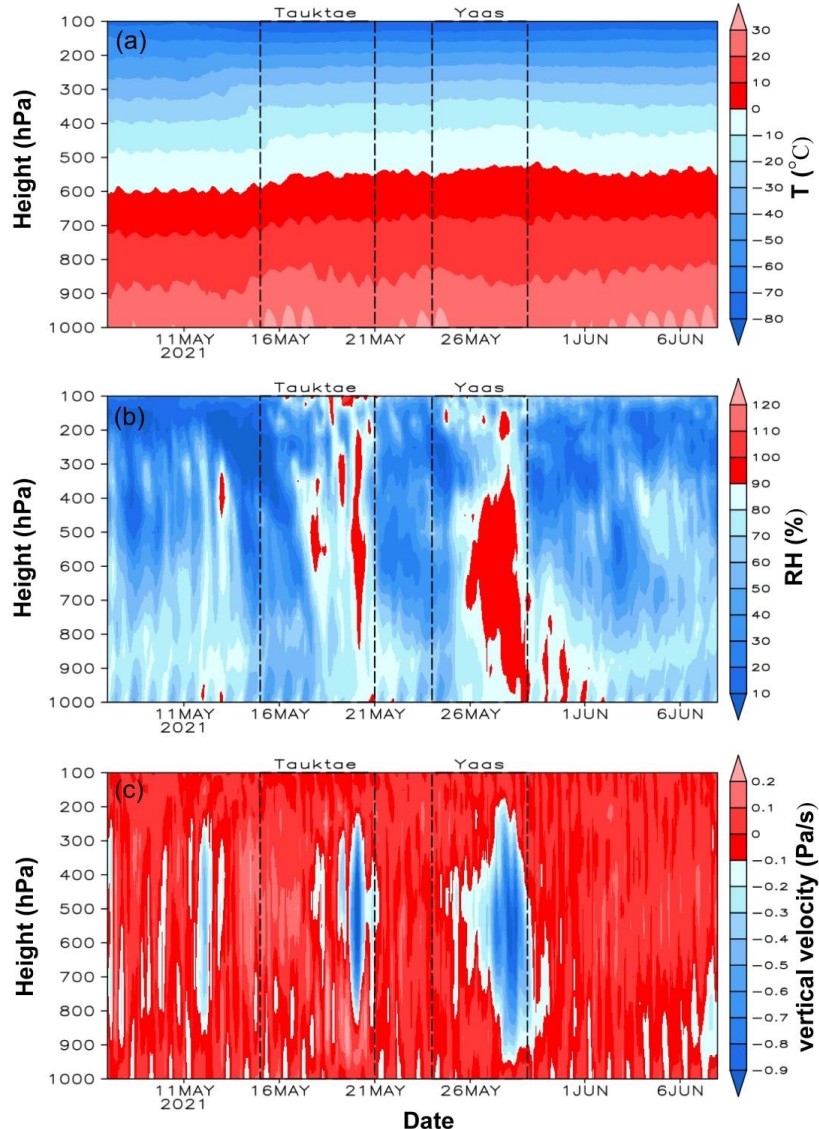


**Figure 8 Time series of the vertical distribution of air temperature (top), RH (middle), and vertical velocity (bottom) averaged over 25° N-28° N and 83° E-87° E with Kathmandu approximately at the centre. Negative (positive) vertical velocities indicate ascending (descending) winds.**







Examining a plot of $\delta D_v$ vs specific humidity, combined with the Rayleigh distillation

and mixing curves, we can assess the mixing conditions during the study period (Figure 9).
Before the development of cyclone Tauktae and during its early stages, the data points lie well
above the mixing curve, suggesting a significant contribution of vapour from local
evapotranspiration. In contrast, during the later stages of Tauktae, $\delta D_v$ was significantly depleted
to levels well below the Rayleigh curve. Similarly, during the early stage of cyclone Yaas, there
are only a few data points between the mixing and Rayleigh curves with the majority well below
the Rayleigh curve, particularly during the late stage of Yaas. These results indicate the
influences of mixing processes and re-evaporation below clouds as described previously
(Galewsky and Samuels-Crow, 2015). After Yaas had dissipated, $\delta D_v$ gradually increased again
with about half of the data points clustered between the mixing and Rayleigh curves and the
remaining data points well above the mixing curve, indicating a strong influence of mixing
processes and locally evaporated vapour, which is also evidenced by the moisture backward
trajectories (Figure 5 lower right panel).



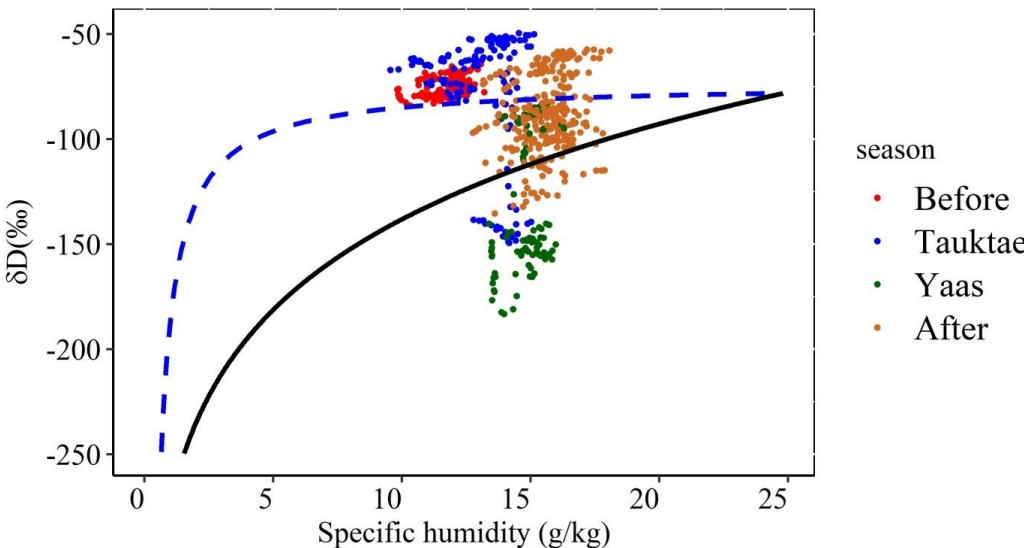


**Figure 9 Scatter plot of hourly averaged δD$_v$ vs. specific humidity (q). The solid black curve**

**represents the Rayleigh distillation curve calculated for the initial condition of δD$_v$ =- 78.20**

**‰, BoB-averaged δD$_v$ (Lekshmy et al., 2022), SST of 30° C, and RH of 90 %. The dashed**

**blue curve represents the mixing line, calculated based on dry continental air (q= 0.5 g/kg**

**and δD$_v$ =-300 ‰ (Wang et al., 2021)) and the wet source, which corresponds to the initial**

**conditions used to calculate the theoretical Rayleigh curve.**

## 3.4   Relationships between local weather parameters and vapour δ$^{18}$O, δD,

### and d-excess

Besides regional influences, we also analyzed whether changes in local meteorological

conditions impact the variations in the isotopic composition of atmospheric water vapour.

Before the cyclone events, both δ$^{18}$O$_v$ and δD$_v$ showed significant negative correlations with

local air temperature and wind speed and significant positive correlations with relative humidity

(Table 2). This correlation between δ$^{18}$O$_v$/δD$_v$ and relative humidity became negative during the



two cyclone events, with a significant temperature effect also present. We hypothesize that the
progressive rainout during the cyclone events followed a temperature decrease (Figure 2), which
would result in this $\delta^{18}O_v/\delta D_v$ correlation with temperature (Delattre et al., 2015). However, the
strength of the correlations between $\delta^{18}O_v/\delta D_v$ and local meteorological parameters varied
significantly throughout the lifetimes of both cyclones. For instance, the effects of temperature
and relative humidity on $\delta^{18}O_v$ were stronger (r=0.68 for temperature and r=-0.74 for RH) during
Yaas compared to Tauktae (r=0.34 for temperature and r=-0.49 for relative humidity). The
weaker relationship during Tauktae is likely due to the significantly lower rainfall amounts
relative to Yaas. The cooling of surface air during rainfall and the associated isotopic equilibrium
of vapour with raindrops cause a positive correlation between $\delta^{18}O_v/\delta D_v$ and temperature
(Midhun et al., 2013). This process was more favourable during Yaas with its stronger and more
continuous rainfall (Fig. 2). During Tauktae, we did not observe any effect of precipitation
amount on the isotopic composition of atmospheric water vapour, while during Yaas there was a
strong negative correlation (r=-0.56) between them. Recent studies have suggested that the
impact of rainfall amount is not a purely local phenomenon (Galewsky et al., 2016) but
modulated by convective and large-scale properties such as downdraft moisture recycling (Risi et
al., 2008), large-scale organized convection and associated stratiform rain (Kurita, 2013), and
regional circulation and shifting moisture source regions (Lawrence et al., 2004). During cyclone
Yaas, our measurements showed the presence of an intense convective system over our study site
which indicates that the observed rainfall amount effect may have been controlled by moisture
convergence (Chakraborty et al., 2016). Subsequent rainfall from the convective system over a
region with depleted isotope values resulted in a negative association between precipitation
amount and $\delta^{18}O_v/\delta D_v$ (Kurita, 2013). Furthermore, the negative correlation between $\delta^{18}O_v/\delta D_v$

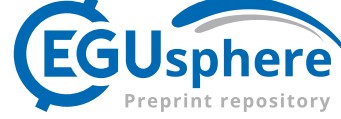

and RH together with the fact that $\delta^{18}O_v/\delta D_v$ was depleted during both cyclone events highlight
the influence of humid moisture sources on the isotopic composition of atmospheric water
vapour (Yu et al., 2008), which was also confirmed by our moisture backward trajectory analysis
(Fig. 5). A strong negative correlation between $\delta D_v$ and RH was also observed in mid-
tropospheric water vapour over the western Pacific associated with intense convective activity
(Noone, 2012). It is noteworthy that the relationship between $\delta^{18}O_v/\delta D_v$ and temperature before
and after the cyclone events degraded significantly, which might be due to the admixture of
vapour originating from plant transpiration during that period (Delattre et al., 2015).

As discussed above, $\delta^{18}O_v$ and $\delta D_v$ were strongly associated with air temperature and RH

during cyclone Yaas but less so during cyclone Tauktae. In contrast, d-excess$_v$ was positively
(negatively) correlated with local air temperature (local RH) before, during Tauktae, and after
both cyclone events, whilst no significant correlations were seen during cyclone Yaas (Table 2).
This indicates that local moisture recycling may have played a crucial role at our sampling site,
while the absence of any correlation of d-excess$_v$ with RH during Yaas implies that RH might
not be a reliable predictor of kinetic fractionation during evaporation at our site. In addition,
while about 75% of RH measurements during Yaas yielded high values (i.e., RH >80%), this
fraction was only 25% during Tauktae. Previous studies (e.g., Midhun et al., 2013; Uemura et al.,
2008) highlighted that the relation between d-excess$_v$ and RH weakens above RH=80%, which
may explain the weaker relation of d-excess$_v$ and RH during Yaas.






**Table 2 Linear correlations between the isotopic composition of atmospheric water vapor**
**($\delta^{18}O_v$, $\delta D_v$, and d-excess$_v$) and air temperature (T), relative humidity (RH), precipitation**
**amount (P), wind speed (WS), and dew point temperature ($T_d$) before, during, and after the**
**cyclone events. \*\*\*,\*\*, and \* indicate correlation significance levels of 0.001, 0.01, and 0.05**
**respectively.**

| | T | RH | P | WS | $T_d$ |
|---|---|---|---|---|---|
| **Before** | | | | | |
| $\delta^{18}O_v$ | -0.34\*\*\* | 0.45\*\*\* | -0.41 | -0.45\*\*\* | 0.16 |
| $\delta D_v$ | -0.1 | 0.28\*\*\* | -0.37 | -0.28\*\*\* | 0.41\*\*\* |
| d-excess$_v$ | 0.68\*\*\* | -0.61\*\*\* | 0.35 | 0.59\*\*\* | 0.40\*\*\* |
| **Cyclone Tauktae** | | | | | |
| $\delta^{18}O_v$ | 0.34\*\*\* | -0.49\*\*\* | 0.11 | 0.20\* | -0.22\*\* |
| $\delta D_v$ | 0.41\*\*\* | -0.55\*\*\* | 0.10 | 0.26\*\* | -0.20\*\* |
| d-excess$_v$ | 0.79\*\*\* | -0.67\*\*\* | -0.22 | 0.75\*\*\* | 0.19\* |
| **Cyclone Yaas** | | | | | |
| $\delta^{18}O_v$ | 0.68\*\*\* | -0.74\*\*\* | -0.56\*\*\* | 0.05 | 0.28\*\* |
| $\delta D_v$ | 0.70\*\*\* | -0.76\*\*\* | -0.56\*\*\* | 0.06 | 0.30\*\* |
| d-excess$_v$ | -0.003 | 0.1 | 0.19 | 0.27\*\* | 0.19\* |
| **After** | | | | | |
| $\delta^{18}O_v$ | 0.13\* | -0.13\* | - | 0.14\* | 0.10 |
| $\delta D_v$ | 0.22\*\*\* | -0.22\*\*\* | - | 0.21\*\*\* | 0.18\*\* |
| d-excess$_v$ | 0.56\*\*\* | -0.54\*\*\* | - | 0.47\*\*\* | 0.47\*\*\* |


## 4   Conclusion


This study presented the results of continuous measurements of the isotopic composition
of atmospheric water vapour over Kathmandu between 7 May and 7 June 2021 covering two
cyclone events, namely cyclone Tauktae formed over the Arabian Sea, and cyclone Yaas formed
over the Bay of Bengal. Both cyclone events led to significant depletion of $\delta^{18}O_v$ and $\delta D_v$, with
$\delta^{18}O_v$ decreasing by over 12 ‰ between May 14 and May 20 (during Tauktae) as well as
between May 24 and May 29 (during Yaas). We could attribute those rapid depletions to changes
in moisture sources (local vs. marine) that were inferred from backward moisture trajectories.



Similar slopes and intercepts of meteoric vapour line with GMWL during both cyclone events
indicate the occurrence of surface recharge following convective conditions. The lower
intercepts before and after the cyclone events highlight the influence of non-equilibrium
processes such as evaporation on the isotopic composition of atmospheric water vapour.

Despite significant diurnal fluctuations in temperature and specific humidity during both

cyclone events, $\delta^{18}O_v$ and $\delta D_v$ exhibit weak diurnal signals which rule out any impact of day-
night variations in local weather parameters. Instead, these discrepancies might reflect different
cyclone sources and convection processes they underwent along their northward trajectories. The
OLR and regional precipitation during cyclone events together with the observed correlation
between vertical velocity and $\delta^{18}O_v$ showed high moisture convergence and heavy convection at
and around the measurement site which caused unusually depleted $\delta^{18}O_v$ during that period.
Moisture convergence and convection were stronger during cyclone Yaas which resulted in
higher (lower) d-excess$_v$ ($\delta^{18}O_v$) values during Yaas, compared to Tauktae, possibly due to
strong downdrafts during the cyclone-related convective rain events which can transport vapour
with higher (lower) d-excess$_v$ ($\delta^{18}O_v$) values toward the surface. During the cyclone events, and
in contrast to immediately before and after these events, there was a strong linear association
between the isotopic compositions of atmospheric water vapour and local meteorological
parameters, which led us to conclude that the progressive rainout during the cyclone events
followed a temperature decrease and RH increase, which would, in turn, produce a $\delta^{18}O_v/\delta D_v$
correlation with temperature and RH. This type of association may visible in the cyclones'
moisture characteristics as each cyclone transported high RH from a remote ocean inland, which
suggested that their specific water vapour stable isotopic signatures could still be observed as far
north as Kathmandu.



Overall, our results showed that tropical cyclones that originated in the BoB and AS

during the pre-monsoon season transported large amounts of isotopically depleted vapour and
produced moderate to heavy rainfall over a sizeable region in Nepal. Hence the isotopic
composition of atmospheric water vapour and precipitation during the dry season should be
interpreted with caution and the effects of cyclones should not be underestimated. Additionally,
our results further underline the need for simultaneous measurements of the isotopic composition
of both atmospheric water vapour and precipitation to better understand post-condensation
exchanges between falling raindrops and boundary layer vapour over Kathmandu.













**Data Availability**

Data will be available upon request from the corresponding author.

**Competing interests**

The contact author has declared that none of the authors has any competing interests.

**Acknowledgements**

This work was funded by 'The Second Tibetan Plateau Scientific Expedition' (Grant No. 2019QZKK0208) and the National Natural Science Foundation of China (Grants 41922002 and 41988101-03). We thank Yulong Yang for his assistance with instrument set-up and initial running.

**Author contributions**

**Niranjan Adhikari**: Data curation, Formal analysis, Writing - Original draft preparation. **Jing Gao**: Data curation, Conceptualization, Methodology, Supervision, Writing - Review and Editing, Funding acquisition. **Aibin Zhao**: measuring assistance, Writing – Editing. **Tianli Xu**, **Manli Chen,** and **Xiaowei Niu**: measuring assistance. **Tandong Yao**: Supervision, Funding acquisition.



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
