# Peer review of "Spring tropical cyclones modulate near-surface isotopic compositions of atmospheric water vapour at Kathmandu, Nepal"

_EGUsphere, 2023_

## Author Comment (AC1)

Responses to Comments on the Manuscript:

**"Spring tropical cyclones modulate near-surface isotopic compositions of atmospheric water vapour at Kathmandu, Nepal"**

**(MS No.: egusphere-2023-2186)**

We thank the two reviewers for their constructive comments. We have made our best to address all the comments. Our point-by-point response in below in blue and reviewers' comments in black.

**RC1**: 'Comment on egusphere-2023-2186', Anonymous Referee #1,

Due to the lacking of high-resolution observations of water vapor isotopic compositions, its evolution pattern and controlling factors remains unclear. In this study, the authors present the continuous observations water vapor isotopes at Kathmandu during the periods of two cyclones in spring formed over the Arabian Sea and the Bay of Bengal. Their results reveal the key processes controlling the isotopic composition of water vapor and have implications for paleoclimate reconstruction of Indian Ocean cyclones. I would like to provide some comments below.

**RC1**: The evolution features of vapor isotopes before, during, and after the two cyclones have been presented in section 3.1. The features of day-to-day isotopic variations presented in section 3.2 are similar to in section 3.1. In addition, the diurnal variations of $\delta^{18}O_v$ and $\delta D_v$ are very weak, which should be overwhelmed by the synoptic processes during the cyclone activity. As a result, I think the section 3.2 is redundant.

**Response:** Thank you for your suggestion. We have integrated essential components of Section 3.2 with Section 3.1 and subsequently omitted the repeated portion. As a result, Section "3.2 Day-to-day and diurnal variations during cyclone events" has been deleted from the revised manuscript. Furthermore, the revised manuscript has been re-arranged with separate sections for results and discussion following the other reviewer's comments.

**RC1:** The authors surmise that the higher d-excess during cyclone Yaas might associated with downdrafts during convective rain events, which is possible. However, the low condensation temperature of precipitation during strong convection and the decrease of relative humidity in the moisture source region also can lead to the increase of d-excess. Can the influences of the latter two factors be excluded? If not, please clarify.

**Response:** Following your constructive comment, we added an in-depth discussion on the impact of convective strength on isotopic depletion, taking cloud-top pressure and temperature (CTP/CTT) as a proxy in section 4.3 from line 581 to 597 as follows:

Given that CTT and CTP are reliable indicators of both moisture convergence and convective strength in prior studies (Cai et al., 2018; Cai and Tian, 2016), we investigate the linear correlation between CTT/CTP (averaged over the 27°N-28°N latitude and 85°E-86°E longitude range, with our site located at the center) and $\delta^{18}O_v$ (Fig. 11). The results demonstrate a weak positive correlation between CTT/CTP and $\delta^{18}O_v$ during cyclone Tauktae, and a robust positive correlation during cyclone Yaas. These correlations exhibit greater strength compared to the correlation observed with local rainfall. Previous research has highlighted positive correlations between $\delta^{18}O$ and CTT/CTP in the East Asian Monsoon suggesting that intense convection and moisture convergence lead to an increase in cloud-top height and a decrease in CTT, causing a reduction in $\delta^{18}O$ (Cai and Tian, 2016). The decrease in $\delta^{18}O_v$ during cyclone Yaas coupled with decrease in CTT and CTP, shows the influence of intensified convective activities and moisture convergence, while the isotopic depletion during cyclone Tauktae is attributed to upstream rainout processes. Furthermore, a negative correlation is evident between d-excess$_v$ and CTT/CTP, with r = -0.52 and r = -0.60 during cyclone Yaas. Conversely, a weak positive correlation is observed during cyclone Tauktae, with r = 0.32 for both CTT and CTP. This relationship implies that lower CTT and CTP during intense convection relate to increased d-excess$_v$ values during the final stage of cyclone Yaas.

**RC1:** The relationship between cyclone isotope values and local weather parameters is of little importance, and changes in isotope values are mainly influenced by moisture sources and strong convective processes. Therefore, section 3.4 could be deleted.

**Response:** Following your suggestion, Section 3.4 has been deleted and a discussion on the general relationship between local meteorological parameters and the isotopic composition of atmospheric water vapor has been incorporated into the results section.

**RC1:** What is indicated by the multiple vertical black dashed lines in Figure 2?

**Response**: The multiple vertical black dashed lines in Figure 2 are just grid lines for X-axis. We have deleted these grid lines in new Figure 3 in the updated manuscript. The revised figure is shown below.

[Figure]

**Figure 3** Water vapour isotopic evolution (hourly averages) before, during, and after the Tauktae and Yaas cyclone events along with associated surface air temperature, relative humidity (RH), and rainfall amount. The cyan dashed line in the figure represents daily variations.

**RC1:** Figures S3-S4 and S7-S8 should be added with the location of the sampling site. In addition, the paths and intensities of the two cyclones need to be presented in more intuitive plots; Figures S3-6 are not enough, although the path of the cyclone center can be seen roughly.

**Response:** Following comments from both reviewers, we have eliminated Figures S3-S4 from the supplementary material. The revised supplementary material now includes the location of the

sampling site on all maps presented. Additionally, we have incorporated the paths and intensities of the two cyclones, along with accumulated rainfall in new Figure 2. See below.

Following your suggestion, we updated Figures S5-S8 into spatiotemporal differences of regional winds, OLR and precipitation amount during the two cyclone events to quantitate variations as new Figures S3.

[Figure]

**Figure 1** Figure showing the intensity and track of cyclone Tauktae (Upper panel) and Yaas (Bottom panel) along with accumulated rainfall during their occurrence.

RC2: 'Comment on egusphere-2023-2186', Anonymous Referee #2,

**General comment**

Adhikari et al. investigated the influence of two pre-monsoon cyclones, Tauktae and Yaas, on the isotopic signal of surface water vapor in the region of the Arabian and the Bay of Bengal. For

that, they measured the isotopic composition of water vapor continuously from 7 May to 7 June 2021 at Kathmandu with a Picarro instrument. They observed strong decreases of d18O and dD in water vapor during both cyclone events. They explained these signals by high moisture convergence and strong convection at and around the measurement site. They conclude that tropical cyclones that originated from the Bay of Bengal and the Arabian Sea modulate isotopic signals of near-surface atmospheric water vapor considerably in Nepal and may have implications for the tropical cyclone activity in the region.

**RC2:** The interest of this paper is the new isotopic data of d18O in surface water vapor at high temporal resolution in Nepal, which is quite rare in this region. I have several points that need to be addressed for an easier understanding of the paper, in terms of paper organization and language. Moreover, I have some concerns about the relatively short length of the record as well as the restriction of the analyses only on surface water vapor, and not on the isotopic composition of precipitation or in higher layers of the atmosphere (through satellite data for example).

**Response:** Thanks for your constructive comments. We carefully checked satellite water isotopic datasets from TES and MUSICA IASI. Both ended before 2021 and we cannot find any available satellite isotopic data match our observation date. We did not find the published isotopic composition of precipitation during our observation date either. However, we used regional winds, OLR, CTT, CTP, and precipitation amount during the two cyclone events to extend the analyses in this manuscript. In addition, the spatial correlation between $\delta^{18}O_v$ and vertical velocity during both events was analysed in the revised manuscript.

**Major specific comments**

**RC2:** The authors focus on two cyclone events and take them together for their analyses (or at least it gives this feeling to the reader), while these two cyclones are quite different in terms of trajectory and precipitation on the measurement site. The authors should clarify the differences and common points of these two cyclones and highlight more these differences for their analyses. Moreover, a figure showing the track and the intensity of the cyclones should be provided to see really the differences between these two cyclones (in section 2.2 for example). Then I expect the authors to more discriminate the effects of the singular cyclone on the isotopic content of water vapor. Maybe different forcing mechanisms between Tauktae and Yaas events are in play (at least the source of water vapor is not the same). Moreover, I think the section 2.4 can be merged to the section 2.2.

**Response:** Following your suggestions, we tried our best to distinguish the controls factors during Tauktae and Yaas. Figures depicting the intensities of the cyclones and accumulated rainfall along the paths, have been added into the revised manuscript as Figure 2. See below for reference.

[Figure]

[Figure]

**Figure 2** Figure showing the intensity and track of cyclone Tauktae (Upper panel) and Yaas (Bottom panel) along with accumulated rainfall during their occurrence.

Additionally, section 2.4 has been merged to section 2.2 in the revised manuscript. We also rewrite the results and discussion parts separately to more discriminate the effects of the singular cyclone on the isotopic content of water vapor in the updated manuscript. In brief, we have discussed the different relationships between $\delta^{18}O_v$ and total rainfall along the moisture trajectories during both cyclone events in Section 4.3 from line 531 to 566. We further examined the distinct roles of convective strength on isotopic depletion, taking cloud-top pressure and temperature (CTP/CTT) as a proxy from line 581 to 597.

**RC2:** In its present form, the paper is not very clear in terms of organization and has some redundancy and not so useful analyses, as already highlighted by the first reviewer. For example, section 3 should be about Results only. After section 3.2, a discussion section 4 about the controlling factor of the isotopic composition of surface water vapor at Kathmandu should be there, with sub-sections for each mechanism. Moreover, an English language editing is probably necessary to make reading easier.

**Response:** Thank you for your constructive comments. As per the comments given by both reviewers, the revise manuscript has been re-arranged with separate sections for results and discussion. We tried our best to clarify and compact the revised manuscript. We introduced the data and methods used in the section 2, described the vapour isotopic variations in section 3 and discussed the possible mechanisms for such variations in section 4. Then conclusions are presented in section 5.

Furthermore, the manuscript has undergone comprehensive revisions and thorough editing to address any English language and grammatical errors by a native speaker.

**RC2:** I am a little bit puzzled by the shortness of the isotopic records. It would be valuable if the authors have a longer record to show the difference of isotopic response to cyclone events and standard precipitation ones (even if they talk about one such event before cyclone Tauktae). Moreover, I think this study in its present from lacks more comprehensive analyses. For example, it would be interesting to investigate the influence of air motion from upper layers in the atmosphere.

**Response:** Following the reviewer's comments, we not only conducted a comparison of $\delta^{18}O_v$, $\delta D_v$, and d-excess$_v$ values during cyclone events relative to a single pre-cyclone event but also extended the analysis by comparing them with the isotopic composition of atmospheric water vapour recorded at the onset of the summer monsoon (i.e., June 2021). The substantial and sustained rainfall observed at the sampling site after the onset of the summer monsoon provides a robust basis for distinguishing the isotopic response to cyclone events with standard precipitation during normal monsoon conditions. The new time series plot of daily average $\delta^{18}O_v$, $\delta D_v$, and d-excess$_v$ observed at Kathmandu site from 07 May to 30 June 2021 together with corresponding surface air temperature, relative humidity, and rainfall amount has been added in supplementary Figure S7 in the updated version. The new figure is displayed below.

[Figure]

**Figure S7** complete time series of daily average $\delta^{18}O_v$, $\delta D_v$, and d-excess$_v$ observed at Kathmandu site in June 2021 together with surface air temperature, relative humidity, and rainfall amount. .

Furthermore, in an effort to comprehend the impact of upstream rainout processes and large-scale atmospheric circulation on the isotopic composition of atmospheric water vapor, we have introduced a new section, Section 4.3, titled "Influence of rainfall." In this section, we distinguish the varying control mechanisms between cyclones Tauktae and Yaas by examining the effects of rainout processes along moisture trajectories, as well as the influence of cloud-top pressure and cloud-top temperature on $\delta^{18}O_v$, $\delta D_v$, and d-excess$_v$. We further added the analysis on spatial distribution of correlation coefficient between $\delta^{18}O_v$ and vertical velocity during both events to comprehend the impact of convective strength on isotopic depletion in each event.

**RC2:** The authors could try to use some satellite isotopic data. If some isotope precipitation data in the region are available, it could be a nice added value for the discussion.

**Response:** We highly value the reviewer's suggestion to incorporate satellite isotopic data. Unfortunately, the satellite isotopic data corresponding to our sampling date is currently unavailable. Specifically, widely-used datasets such as TES only provide isotopic data for tropospheric water vapour up to the year 2018 and MUSICA IASI provide data up to the year 2019. Despite our efforts, we were unable to locate precipitation and upstream vapor isotopic data with a date closely aligned to our measurements. However, we used regional winds, OLR, CTT, CTP, vertical velocity, and precipitation amount during the two cyclone events to extend the analyses in this manuscript.

**RC2:** I do not understand why the authors use spatially coarse NCEP data for the outgoing longwave radiation (OLR). Like for the other fields like winds…, the authors should use the state-of-the-art ERA5 data.

**Response:** Following the reviewer's comment, we have used the state-of-the-art ERA5 data for OLR as shown below. This dataset provides hourly OLR with spatial resolution of 0.25°×0.25°. The results align closely with our original findings.

[Figure]

**Figure 3** Time series of daily averaged d-excess$_v$ (a), $\delta^{18}O_v$ (b), and Hovmöller diagram of OLR (W/m$^2$) averaged over 80° E-90° E (c) The solid parallel lines in the (c) depict the latitude range of sampling site.

**RC2:** Data availability: the data should be made openly available in a public repository like Zenodo, Figshare, PANGAEA… once the paper accepted (if it is the case). This extremely important point is a sufficient reason for rejecting a paper, especially in open-access EGU journals.

**Response:** We agree with the reviewer. The data used in this study will be openly available in the Zenodo after the paper is accepted.

**Technical comments:**

**RC2:** Line14: please rephrase.

**Response:** This sentence has been rephrased in the revise manuscript.

**RC2:** Lines 16-18: the isotopes are a tool, not an end in themselves. In other words: why is it interesting to investigate the impact of cyclones on the isotopic composition of water vapor?

**Response:** This sentence has been deleted and the abstract has been modified accordingly.

**RC2:** Line 28: during cyclone Yaas, which resulted…

**Response:** After the thorough revision of the manuscript, this sentence has been eliminated in the revision.

**RC2:** Lines 45-46: season, also known as the pre-monsoon season.

**Response:** This sentence has been corrected into "…agricultural activities also rely on precipitation in the pre-monsoon season." in the revision.

**RC2:** Lines 48-49: already said just before.

**Response:** The repeated sentence has been deleted in the revise manuscript.

**RC2:** Line 57: replace "while" by "and".

**Response:** This sentence has been corrected into "According to the International Best Track Archive for Climate Stewardship (IBTrACS) project, in 2019 three cyclones originated in the BoB and five cyclones originated in the AS." in the revision.

**RC2:** Line 116: replace "during" by "including".

**Response:** This sentence has been corrected in the revision.

**RC2:** Lines 125-126: give a reference for the temperature and rainfall records.

**Response:** Reference of the "Department of Hydrology and Meteorology, Government of Nepal" has been added in the revision.

**RC2:** Line 132: specific humidity of where? At the surface, at 1000 hPa, vertically integrated?

**Response:** Thank you for your comment. Here we have used the vertically integrated specific humidity (averaged over 1000 hPa to 850 hPa). The sentences have been modified as "…a substantial presence of moisture was observed over extensive areas encompassing the BoB, the AS, India, and surrounding regions including our sampling site. Figure 1 shows the elevated specific humidity levels averaged between 1000 hPa and 850 hPa throughout the duration of our study period…."

**RC2:** Figure 1: First, I am not sure if the plot (a) is so useful. Second, the quality of the figure needs to be improved, especially the plot (b). Third, I would let only the specific humidity in background colors (brown to green colormap for example) and remove the mean vertically

integrated moisture flux to improve the readability of the figure. You could put a bigger dot in red for the location of the site.

**Response:** Following the reviewer's comments, Figure 1a has been deleted in the revised manuscript and Figure 1b has been modified. The mean vertically integrated moisture flux has been removed and specific humidity has been plotted in color for more readability in the figure below.

[Figure]

**Figure 4** Spatial distribution of specific humidity averaged over 1000 hPa to 850 hPa (in g/Kg) during the period of study. The yellow dot shows the location of    Kathmandu.

**RC2:** Line 143-144: specific humidity at the surface?

**Response:** Figure caption has been modified into "Spatial distribution of specific humidity averaged over 1000 hPa to 850 hPa (in g/Kg) during the period of study. The yellow dot shows the location of Kathmandu." in the revise manuscript.

**RC2:** Section 2.2: see my major comment 1.

**Response:** Modification has been made following the major comment 1.

**RC2:** Section 2.4: it should be merged with section 2.2.

**Response:** Following your suggestion, section 2.4 has been merged with section 2.2 in the revision.

**RC2:** Lines 205-206: … tropical cyclones (Jackisch et al., 2022). They have been used…

**Response:** This sentence has been corrected in the revised manuscript.

**RC2:** Line 214: please add the reference of the ERA5 data (Herbash et al., 2020). See the reference section.

**Response:** Correct reference (Herbash et al., 2020) has been added in the revised manuscript.

**RC2:** Lines 222-227: I do not know how to use HYSPLIT, but it sounds the parameters were set just like this. The starting height for the moisture backward trajectories is set to 500 m above ground, and then you say that the average boundary layer height at Katmandu is about 620 m. So why not setting 620 m as an initial starting height and is it important? Please comment.

**Response:** Considering the variation in boundary layer height at Kathmandu during the study period, ranging from approximately 100 m to 1170 m, and with the majority of the data falling below 600 m, we set the initial starting height for the moisture backward trajectories to 500 m above ground. Furthermore, we did not find any significant differences between trajectories at 500 m and 620 m. Therefore, taking 500 m for an initial starting height seems appropriate to derive the moisture trajectories during the period of our study. The moisture backward trajectories plotted at 620 m as an initial starting height is shown below.

[Figure]

**Figure 5** Five-day backward moisture trajectories reaching the sampling site before, during, and after the cyclone events. Colours denote specific humidity (q in g/kg) along the trajectories.

**RC2:** Figure 2: The figure needs to be improved. The background colors should be more transparent for a better readability. The contrast between time series of hourly and daily variations should be strengthened.

**Response:** The figure has been modified in the revise manuscript with more transparent background colors. Additionally, the vertical grid lines have been removed from the figure as pointed out by reviewer 1.

**RC2:** Line 235: replace "surrounding" by 'before and after" (or something similar).

**Response:** This sentence has been corrected in the revision.

**RC2:** Lines 255-257: Because there was not enough time for the d18O of water vapor to recover from the first cyclone.

**Response:** Correction has been made accordingly in the revision.

**RC2:** Section 3.2 and figure 4: For the moment, I do not see the interest of this section. Moreover, did you remove the long-term (i.e., synoptic) trend before doing the muti-daily mean? And an average over 4 days only is quite short anyway.

**Response:** Following both reviewers' suggestions, we removed the diurnal variation described in section 3.2 in the revise manuscript.

**RC2:** From section 3.3: see major comment 2. Please reorganize the paper with clearly separated results and discussion sections. Or it can be by forcing mechanism.

**Response:** Many thanks for such constructive comments. The revised manuscript has been re-organized with separate result and discussion sections.

**RC2:** Lines 383-385: this part is interesting and should be highlighted.

**Response:** Thank you for your comment. The findings presented in lines 383-385 have been further explained within the context of isotopic response, specifically in relation to cyclone events and normal monsoon rain. For additional information, please refer to the response provided in major comment 3.

**RC2:** Line 398: what are the in situ measurements? Please add a reference and a description in the method section.

**Response:** We have mentioned all the in situ measurements in data and methods section and necessary reference has been added.

**RC2:** Line 471: our study site (Fig. S9).

**Response:** This sentence has been corrected in the revision.

**RC2:** Lines 479-482: see major comment 1. Maybe the authors could make one section for each cyclone…

**Response:** We re-organized the manuscript and please refer to the response of major comment 1.

**RC2:** Lines 488-489: intensity weakened, both d18Ov and dDv started to increase again (Fig. 2) like due to…

**Response:** This sentence has been corrected in the revision.

**RC2:** Line 497: during our period of study.

**Response:** This sentence has been corrected in the revision.

**RC2:** Line 601: may be visible…

**Response:** This sentence has been deleted in the revised manuscript.

**RC2:** Lines 609-612: see major comment 3. It's also interesting to see how the isotopic content of upstream moisture, through transport and convection, influenced the isotopic content of precipitation at the site.

**Response:** We agree with the reviewer. Regrettably, we are unable to analyze the isotopic content of upstream moisture due to the lack of corresponding vapour and precipitation isotopic data for our sampling date. Additional details can be found in the response to major comment 3.

**RC2:** Figures S3 and S4: change the colormap for considering color vision deficiencies. Also, make the land coast thicker.

**Response:** After detailed revision of the manuscript, we have eliminated Figures S3-S4 from the supplementary material. All maps in the revised supplementary material now include the location of the sampling and the land coast is also thicker than before.

**RC2:** Figures S5 and S6: make the land coasts thicker.

**Response:** Correction has been made in the revision.

**RC2:** All maps in supplementary material: show the Kathmandu site.

**Response:** The Kathmandu site has been shown in all maps in the supplementary material.

---

## Editor Decision (ED1)

**Editor comments on Manuscript No egusphere-2023-2186**

Spring tropical cyclones modulate near-surface isotopic compositions of atmospheric water vapour at Kathmandu, Nepal by Adhikari et al.

I agree with the referee that the manuscript has significantly improved. However, I have several comments that should be considered before the manuscript can be accepted for publication in ACP.

**Major comment:**

My major concern is that the structuring of the result and discussion section is not adequate for ACP yet. I know that you separated the sections due to the comment of one referee. However, the discussion section as it is now is much too long and contains too much that rather belongs to the result section. So either, you go back to what you had before that you combine these sections and call it results and discussion or you move major parts to the result section and shorten the discussion to max. 3 pages.

**Minor comments and technical corrections:**

P5, L99: I am not sure if the correction of "during" to "including" is really correct. Do you evaluate the isotopic composition "during" a pre-monsoon cyclone event or do you have that additionally. In the latter case it would be indeed "including", in the former however it would be rather "during". Please clarify and change sentence accordingly.

P5, L101: These cyclones did not pass Kathmandu, but how far away where these and what did you see in the measurements? I understood that you saw them in the measurements, so that means you still have remnants or an influence of these cyclones over Kathmandu.

P5, L109: Please clarify and change text accordingly over which time period this precipitation amount given is referring to. Is it over one month or one day or per year?

P6, L112: ranges -> ranging and change "averaged" to "is averaged".

P6, L118: Please mention instead of just "study period" explicitly which time period is considered.

P6, Figure 1: I think this figure could rather be moved to the Appendix.

P9, Figure 2: Also here, please give the time period over which the accumulated rainfall is given. Additionally in the caption the source of the data should be added.

P11, L190-191: GPM appears here twice. I guess once it is obsolete.

P11, L191: Add "for" before "latitude".

P11, L199: Check grammar.

P11, L203: of -> for

P13, Figure 3: In the legend spaces between parameter and unit should be added.

P14, Figure 3 caption, L219: add "as indicated by the color shading".

P14, L220: Rather "average" than "variations"?

P14, L223: Add "in isotopic composition" so that it reads "depletion in isotopic composition".

P14, L232: Change "ranges from" to  "the range was from".

P14, L235: Add "to" so that it reads "to recover pre-cyclone values".

P14, L237: Add "a" so that it reads "a d-excess….."

P15, L240: Here you refer to Fig. S3, but Fig. S1 and Fig. S2 have not been mentioned yet.

P15, L238 ff: Since here and also in Sect. 4 Fig. S3 and S4 are discussed here in detail these should be rather appear in this section than in the Appendix.

P19, L298: Discussions -> Discussion. As mentioned before this section is much too long for a discussion and the majority of the content rather belongs to the result section.

P19, L300: back trajectories -> backward trajectories

P19, L303: I would change here "depletion" to "composition". The depletion is the result of the cyclones, but what you are exploring is the composition.

P20, L318: majority of AS vapour -> not clear, please rephrase

P21, Figure 5 caption: Delete "moisture" and add "along the trajectories" after specific humidity.

P24, Figure 6 caption and text: Units should be given in the following format: g kg-1 (see ACP guidelines).

P25, L400: As mentioned before, since Fig. S3 and S4 seem to play a larger role for your study than just some additional information these should rather appear in the manuscript itself than in the appendix.

P25, L407: Check reference to the figures. Fig. S5 and S6 show precipitation

P27, L446ff: Here your results/discussion are based mostly on Fig. 3 and Fig. 8 is only mentioned once. What is the purpose of Fig. 8? If this figure is not that important it should be moved to the Appendix.

P34, Figure 10. Also this figure could be rather moved to the Appendix since it is not really discussed/mentioned in the manuscript.

P33, L528ff: This last subsection is quite long and not solely about rainfall. It seems that there is some subsection header missing.

P33, L578: Should here the new subsection start? These parts are not about rainfall. Further, what is "CTT" and "CTP"? Abbreviations should be introduced.

P38, L619: Not clear, please rephrase what exactly what you mean with "unlike during Yaas".

---

## Author Response (AR2)

**Response to the editor**

 **Editor comments on Manuscript No egusphere-2023-1371**

Spring tropical cyclones modulate near-surface isotopic compositions of atmospheric water vapour at Kathmandu, Nepal by Adhikari et al.

I agree with the referee that the manuscript has significantly improved. However, I have several comments that should be considered before the manuscript can be accepted for publication in ACP.

**Response: Many thanks for your constructive comments. Below are our point-to-point responses to the comments. The comments are in black, and our responses are in blue.**

**Major comment:**

My major concern is that the structuring of the result and discussion section is not adequate for ACP yet. I know that you separated the sections due to the comment of one referee. However, the discussion section as it is now is much too long and contains too much that rather belongs to the result section. So either, you go back to what you had before that you combine these sections and call it results and discussion or you move major parts to the result section and shorten the discussion to max. 3 pages.

Response: Thank you for your suggestions. Following a comprehensive review of the manuscript, we have decided to maintain the original structure with "Results and discussion" as a combined section, as initially formatted.

**Minor comments and technical corrections:**

P5, L99: I am not sure if the correction of "during" to "including" is really correct. Do you evaluate the isotopic composition "during" a pre-monsoon cyclone event or do you have that additionally. In the latter case it would be indeed "including", in the former however it would be rather "during". Please clarify and change sentence accordingly.

Response: Since, we evaluate the isotopic composition of atmospheric water vapour during the cyclone events, we replaced "including" by "during" in the revision.

P5, L101: These cyclones did not pass Kathmandu, but how far away where these and what did you see in the measurements? I understood that you saw them in the measurements, so that means you still have remnants or an influence of these cyclones over Kathmandu.

Response: Yes, we saw the influence of these cyclones over Kathmandu by our measurements. To avoid misunderstanding to readers, we deleted this sentence and modified into: "A substantial influence of these cyclone events on the sampling site for several days is apparent in the isotopic composition of atmospheric water vapour, showcasing a marked depletion in comparison to normal days." We also mentioned in section 3.3 in Line 533 that "Yaas came as close as 330 km to our site, while Tauktae was 1050 km away when it dissipated (Fig. S9)."

P5, L109: Please clarify and change text accordingly over which time period this precipitation amount given is referring to. Is it over one month or one day or per year?

Response: Thank you for your comment. We have now included "average annual precipitation" in the revision.

P6, L112: ranges -> ranging and change "averaged" to "is averaged".

Response: Correction has been made in the revised manuscript.

P6, L118: Please mention instead of just "study period" explicitly which time period is considered.

Response: We have now included exact date of study period in the manuscript instead of "study period".

P6, Figure 1: I think this figure could rather be moved to the Appendix.

Response: Following your comment, Figure 1 has been moved to Supplementary material as Figure S1.

P9, Figure 2: Also here, please give the time period over which the accumulated rainfall is given. Additionally in the caption the source of the data should be added.

Response: The time period over which the accumulated rainfall was calculated has been mentioned and the data source has also been included in the caption.

P11, L190-191: GPM appears here twice. I guess once it is obsolete.

Response: We have corrected it.

P11, L191: Add "for" before "latitude".

Response: Correction has been made in the revision.

P11, L199: Check grammar.

Response: The whole paragraph has been modified to "We further acquired data on outgoing longwave radiation (OLR), zonal and meridional winds, specific humidity, vertical velocity, pressure, vertical distribution of relative humidity and temperature from ERA5 datasets (Herbash et al., 2020). The data has a spatial resolution of 0.25° based on longitude-latitude grids"

P11, L203: of -> for

Response: Correction has been made accordingly.

P13, Figure 3: In the legend spaces between parameter and unit should be added.

Response: Following your suggestions, we have added spaces between parameters and units in all necessary figures in the revised manuscript.

P14, Figure 3 caption, L219: add "as indicated by the color shading".

Response: Modification has been made.

P14, L220: Rather "average" than "variations"?

Response: In the caption, we have changed "variations" into "averages".

P14, L223: Add "in isotopic composition" so that it reads "depletion in isotopic composition".

Response: Modification has been made following your suggestion.

P14, L232: Change "ranges from" to "the range was from".

Response: Modification has been made following your suggestion.

P14, L235: Add "to" so that it reads "to recover pre-cyclone values".

Response: Modification has been made

P14, L237: Add "a" so that it reads "a d-excess….."

Response: We have added "a" before d-excess in the revised version.

P15, L240: Here you refer to Fig. S3, but Fig. S1 and Fig. S2 have not been mentioned yet.

Response: We are extremely sorry for this mistake. We have now mentioned Figure S1 and S2 in the revised manuscript.

P15, L238 ff: Since here and also in Sect. 4 Fig. S3 and S4 are discussed here in detail these should be rather appear in this section than in the Appendix.

Response: We agree with the editor. Now Figure S3 and S4 have been moved to section 3.3 in the revised manuscript.

P19, L298: Discussions -> Discussion. As mentioned before this section is much too long for a discussion and the majority of the content rather belongs to the result section.

Response: Following the major comment, we have once again restructured the manuscript, now incorporating both results and discussion within a single section titled "Results and discussion."

P19, L300: back trajectories -> backward trajectories

Response: Change has been made.

P19, L303: I would change here "depletion" to "composition". The depletion is the result of the cyclones, but what you are exploring is the composition.

Response: Following your comment, we have changed "depletion" to "composition" in the revised manuscript.

P20, L318: majority of AS vapour -> not clear, please rephrase

Response: The sentence has been modified to "As cyclone Tauktae approached the continent, the primary moisture to Kathmandu was coming from the Arabian Sea, instead of local origins."

P21, Figure 5 caption: Delete "moisture" and add "along the trajectories" after specific humidity.

Response: Modification has been made accordingly.

P24, Figure 6 caption and text: Units should be given in the following format: g kg-1 (see ACP guidelines).

Response: Correction has been made in all necessary figures in the manuscript.

P25, L400: As mentioned before, since Fig. S3 and S4 seem to play a larger role for your study than just some additional information these should rather appear in the manuscript itself than in the appendix.

Response: As mentioned above, Figure S3 and S4 have been moved to section 3.3 in the revised manuscript

P25, L407: Check reference to the figures. Fig. S5 and S6 show precipitation

Response: Correction has been made.

P27, L446ff: Here your results/discussion are based mostly on Fig. 3 and Fig. 8 is only mentioned once. What is the purpose of Fig. 8? If this figure is not that important it should be moved to the Appendix.

Response: After a thorough revision of the manuscript, we have recognized that Figure 8 holds less significance, whereas Figure S8 is deemed more important. Consequently, we have relocated Figure 8 to the supplementary material, and Figure S8 has been moved to section 3.3 in the revised manuscript.

P34, Figure 10. Also this figure could be rather moved to the Appendix since it is not really discussed/mentioned in the manuscript.

Response: Figure 10 has been moved to supplementary material.

P33, L528ff: This last subsection is quite long and not solely about rainfall. It seems that there is some subsection header missing.

Response: The new subsection has been introduced here as "3.5 Relation with cloud-top temperature and cloud-top pressure".

P33, L578: Should here the new subsection start? These parts are not about rainfall. Further, what is "CTT" and "CTP"? Abbreviations should be introduced.

Response: We have introduced new sub-section "3.5 Relation with cloud-top temperature and cloud-top pressure" here and abbreviations for CTT and CTP have been introduced.

P38, L619: Not clear, please rephrase what exactly what you mean with "unlike during Yaas".

Response: Upon careful examination, we have determined that the phrase "unlike during Yaas" can be omitted.

**Response to the referee**

I thank the authors for considering the comments from the two reviewers. The paper now reads better. Here are some minor technical comments to consider before publication:

Response: We express our sincere gratitude to the reviewers for their insightful feedback, which greatly contributed to the substantial improvement of our manuscript.

- Figure 1: put a big black thick cross (for example) instead of the yellow dot.

Response: Figure 1 has been modified accordingly in the revised manuscript as the new Figure S1 following the editor's comments.

- Figure 3: make the curve of daily variations (cyan) thicker and in dark blue for example. I's hard to see now. Also, increase the font size of the text "before", Tauktae"...

Response: We've improved the figure by thickening daily variation curves, changing the color to blue, and increasing the font size for enhanced clarity.

- Could Figures 3 and S7 be merged?

Response: We appreciate the reviewer's suggestion, and after careful consideration, we acknowledge the focus of our manuscript on the isotopic composition of atmospheric water vapor during two spring cyclones. The entirety of our analysis revolves around these cyclone events. Combining both figures into the main body of the manuscript poses a challenge, as it would extend the study period and require the inclusion of additional analysis which is beyond the primary aim of our paper.

Furthermore, we note that Figure S7 is specifically addressed in section 4.2 (section 3.3 in the revised manuscript), where we compare $\delta^{18}O_v$, $\delta D_v$, and d-excess$_v$ during both cyclone events and normal monsoon onset periods. To maintain the manuscript's primary purpose, we believe it is essential not to merge Figure S7, as its inclusion would divert from the main focus of our research.

---

## Author Response (AR3)

I am pleased to inform you that your manuscript is accepted for publication after consideration of the following technical corrections:

Response: We extend our sincere gratitude to editor and anonymous reviewers for the invaluable role you played in the review process of our paper. The insightful comments and constructive feedback significantly contributed to enhancing the quality of the manuscript.

P6, L123: 850 -> unit is missing. I guess you mean hre 850 hPa.

Response: Correction has been made.

P11, L207: Herbash -> Hersbach

Response: Correction has been made.

P16, L283+287: When you write what LWML stands for you start with captial letters, but for GWML everything is written with small letters. It should be the one or toher way for both.

Response: We have made uniform representation for Local Meteoric Vapour Line and Global Meteoric Water Line.

P24, L399: Error message that the reference is not found shows up.

Response: Correct figure number has been cited in the manuscript.

P29, L465: Figure 88b -> Figure 8b

Response: Correction has been made.

P30, L493: Error message that the reference is not found shows up here too.

Response: Correct Figure number has been cited in the manuscript.

P30, L490: Figure 99b -> Figure 9b

Response: Correction has been made.

P31, L502: Figure 99c -> Figure 9c.

Response: Correction has been made.